# MARS: A NEUROSYMBOLIC APPROACH FOR INTERPRETABLE DRUG DISCOVERY

## ABSTRACT

Neurosymbolic (NeSy) artificial intelligence describes the combination of logic or rule-based techniques with neural networks. Compared to neural approaches, NeSy methods often possess enhanced interpretability, which is particularly promising for biomedical applications like drug discovery. However, since interpretability is broadly defined, there are no clear guidelines for assessing the biological plausibility of model interpretations. To assess interpretability in the context of drug discovery, we devise a novel prediction task, called drug mechanism-of-action (MoA) deconvolution, with an associated, tailored knowledge graph (KG), *MoA-net*. We then develop the *MoA Retrieval System (MARS)*, a NeSy approach for drug discovery which leverages logical rules with *learned* rule weights. Using this interpretable feature alongside domain knowledge, we find that MARS and other NeSy approaches on KGs are susceptible to reasoning shortcuts, in which the prediction of true labels is driven by "degree-bias" rather than the domain-based rules. Subsequently, we demonstrate ways to identify and mitigate this. Thereafter, MARS achieves performance on par with current state-of-the-art models while producing model interpretations aligned with known MoAs.

## 1 INTRODUCTION

Drug discovery (DD), the search for novel drugs or chemical compounds to treat ailments, often involves the screening of thousands of small compounds (Lin et al., 2020). Many computational approaches have been developed to accelerate and streamline this screening process (Gottlieb et al., 2011; Gan et al., 2023). Specifically, hundreds of such approaches operate upon knowledge graphs (KGs), in which nodes representing drugs, proteins, or medical conditions are connected by edges, representing the relationships between them (Chen et al., 2020). Typically, DD is formulated on a KG as a link prediction task between drugs and the corresponding medical conditions (indications) to be treated (Schultz et al., 2021; Rivas-Barragan et al., 2022).

It is also important to understand each drug's mechanism-of-action (MoA), the molecular processes by which it achieves its medicinal effect. For instance, as depicted in Fig. 1, MoAs typically involve chains or paths of physical, molecular interactions induced by a drug (Crino, 2016). Uncovering these interactions informs researchers as to how each drug works and de-risks potential side effects (Palve et al., 2021; Green et al., 2023). Revealing MoAs alongside computational DD, a task we call *MoA deconvolution*, requires model *interpretability*: transparency into the processes or patterns which led to certain predictions (Molnar, 2022). Unfortunately, most state-of-the-art techniques on KGs rely on "black-box" models (Wu et al., 2020). Recently, neurosymbolic (NeSy) approaches, which combine logical rules with neural networks (DeLong et al., 2024) have been positioned as a promising avenue for MoA deconvolution because they tend to possess enhanced interpretability.

However, interpretability is broadly defined (Molnar, 2022), which poses an additional challenge: there are no clear guidelines for assessing the plausibility of model interpretations, especially for this novel task. Although some previous studies present explainable or interpretable pipelines (Rivas-Barragan et al., 2022; Urbina et al., 2021), the corresponding explanations leverage *associative* patterns: two nodes with mutual connections are likely to share other connections (Paul et al., 2021). For example, such methods utilize associations regarding a drug's pharmacological class (Ratajczak et al., 2022), side effects (Liu et al., 2021), or known indications

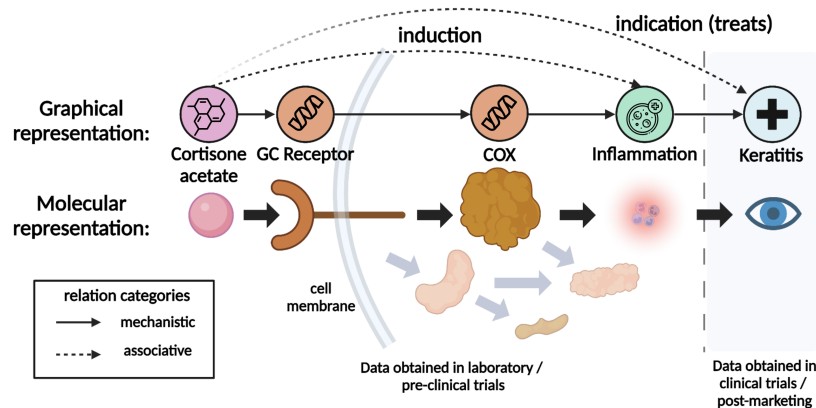

Figure 1: MoA of cortisone acetate. Cortisone acetate upregulates the activity of the glucocorticoid (GC) receptor protein, which, in turn, downregulates the cyclooxygenase (COX) protein. Since COX is directly involved in creating inflammation, its inhibition reduces inflammation, thereby treating keratitis (Gonzalez-Cavazos et al., 2023). Data regarding protein interactions and biological processes (left) can be collected in a laboratory setting, whereas physiological effects like indications (right) are obtained during or after clinical trials.

(Fernández-Torras et al., 2022). Unfortunately, these *associative* patterns, which are discovered during or after clinical trials, are rare or absent for *novel* compounds. Furthermore, such patterns can not represent the MoA of a drug; instead, an MoA involves *mechanistic* patterns, such as the physical, molecular interactions shown in Fig. 1 (Gonzalez-Cavazos et al., 2023). Therefore, within this study, we focus upon model interpretability which provides mechanistic insight into drug MoAs.

We propose *MoA deconvolution*, the prediction of mechanistic paths between drugs and their biological effects, as a prediction task for evaluating *interpretable* methods on KGs. To benchmark this task, we generate a tailored KG, called *MoA-net*, from real-world, experimental data. To perform MoA deconvolution, we introduce a NeSy DD approach called the MoA Retrieval System (MARS). To create MARS, we draw inspiration from previous NeSy methods (Liu et al., 2021; Drancé et al., 2021). However, unlike its predecessors, MARS achieves enhanced interpretability by learning weights associated with logical rules which resemble MoAs. Following training, rule weights reflect relative usefulness to MARS' reasoning processes.

Alongside biomedical domain knowledge, MARS' enhanced interpretability reveals a reasoning shortcut in which predictions are based on unintended semantics (Marconato et al., 2024b). Essentially, predictions upon *MoA-net* are driven by "degree-bias", an artifact of node degree variance (Zietz et al., 2024), rather than the rules representing domain knowledge. Therefore, to address this, we consider the desiderata from Marconato et al. (2024a) for making NeSy systems *shortcut-aware*: (1) *calibration*, high accuracy on concepts unaffected by reasoning shortcuts, (2) *performance*, high accuracy despite reasoning shortcuts being present, and (3) *cost effectiveness* achieved through simple mitigation strategies. Using these desiderata as guidelines, we make MARS shortcut-aware for more insightful predictions involving DD and MoA deconvolution. Ultimately, our study underscores the importance of evaluating the capabilities of NeSy models within applied domains: by evaluating model interpretations against specific domain knowledge, we can more easily identify and mitigate shortcuts.

## 2 RELATED WORK AND BACKGROUND

**NeSy AI for MoA deconvolution**. Several NeSy approaches involve logical rules reflecting path-like patterns in biomedical KGs. For example, Sudhahar et al. (2024) investigates *evidence chains*, paths explaining associations between drugs and diseases. However, these explanations are derived separately from indication predictions, using an additional rule-mining model

(Meilicke et al., 2019). Other approaches (Liu et al., 2021; Drancé et al., 2021) accomplish similar tasks through deep reinforcement learning (RL), in which a neural network contributes toward the optimization of a reward function (Acharya et al., 2023). In these specific cases, reasoning is *guided* by the path-like rules, and predictions are expected to align, to some extent, with such rules. In contrast to these previous studies, we focus upon paths representing MoAs, involving mechanistic, molecular relations. In this study, we also identify a major risk: the approach may neglect to utilize rules in favor of *other* semantics for reward optimization. This results in *reasoning shortcuts*.

**Reasoning Shortcuts**. Several NeSy approaches are designed to abide by rules and domain knowledge (Drancé et al., 2021; Dash & Goncalves, 2021), which might portray such approaches as more trustworthy than neural, black box ones (Gaur & Sheth, 2024). Recent studies, however, have found that NeSy approaches may suffer from reasoning shortcuts, in which a model predicts the correct outcome via unintended semantics (Marconato et al., 2024b; Li et al., 2024b). While reasoning shortcuts are not exclusive to NeSy methods (Jiang & Bansal, 2019; Li et al., 2024a), they may be more easily overlooked when such approaches are portrayed as trustworthy.

## 3 MARS: A NeSy approach for MoA deconvolution

Here, we build the MoA Retrieval System (MARS) to perform MoA deconvolution. MARS improves upon a method called Policy-guided walks with logical rules (PoLo) (Liu et al., 2021) by introducing dynamic, *learned* rule weights. This differs from previous approaches, where weights are static and pre-computed (*e.g.,* mined or literature-derived) (Liu et al., 2021; Drancé et al., 2021). As discussed further, these learned weights also make MARS shortcut-aware.

As shown in Fig. 2, MARS takes two major inputs. The *first* involves a KG. A KG uses nodes to represent entities and edges between them to represent relationships. A KG *triple* comprises two nodes connected by an edge of some specific type, or *relation*. Here, we represent triples as binary predicates: for example, the binary predicate interacts($Protein, Protein$) states that two $Protein$ nodes are connected via the interacts relation. *Node degree* describes the number of edges connected to a node.

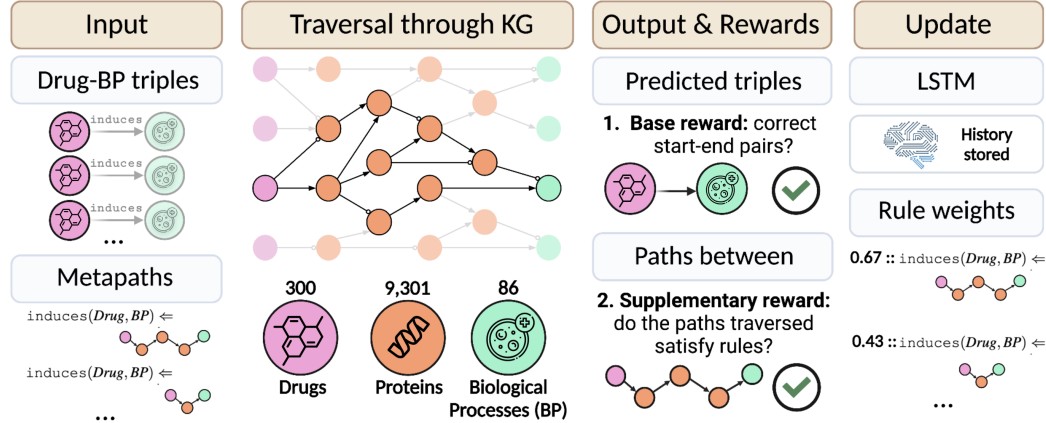

Figure 2: Overview of the MoA retrieval system (MARS).

Specifically, the input KG must contain triples involving some relation of interest. As shown in Fig. 1, some information, such as indications, are discovered *during* or *after* clinical trials, so this information is typically unavailable for novel compounds. Therefore, to understand each MoA as the *biological* response to drug administration, we aim to investigate relations between drugs and biological processes (BPs), such as *signal transduction* or *inflammation* (Consortium, 2019) (*i.e.,* induces($Drug, BP$)). Specifically, we accomplish this through a *link prediction* task, in which we predict whether edges of type induces exist between $Drug$ and $BP$ nodes. Thereafter, new predictions regarding the induces relation serve as potential therapeutic outcomes for the chemical compound represented by the $Drug$ node. As per our knowledge, this is a novel application in

the KG field. Further details on our KG are introduced within section 3.4. The *second* input, as depicted in Fig. 2, includes metapaths of the KG with corresponding weights.

## 3.1 METAPATH-BASED RULES

Metaphs are abstract representations of instantiated paths in a graph (Sun et al., 2011; Himmelstein et al., 2017; Noori et al., 2023). For example, given the following path, $P$, in our KG:

$$Cortisone\ acetate \xrightarrow{upregulates} GC\ receptor \xrightarrow{interacts} COX\ protein \xrightarrow{participates} Inflammation$$

the corresponding metapath, $\tilde{P}$, would be:

$$Drug \xrightarrow{upregulates} Protein \xrightarrow{interacts} Protein \xrightarrow{participates} Biological\ Process$$

Within this study, metapaths can be understood as a sequence of triples within the KG structure, making them inherently interpretable. In MARS, metapaths are used as the bodies of logical rules, in which triples are connected by logical conjunctions ($\wedge$). Conjunctions indicate that, if all triples in the rule body are true, then the rule *head* is evaluated as true. The rule *head*, the left side of the implication arrow ($\Leftarrow$), is a single triple representing the relation of interest between the first and last node types of the metapath, *e.g.*,:

$$\texttt{induces}(Drug, BP) \Leftarrow \texttt{upregulates}(Drug, Protein_A) \wedge$$
$$\texttt{interacts}(Protein_A, Protein_B) \wedge \texttt{participates}(Protein_B, BP)$$

If a rule head is evaluated as true, then the rule is *satisfied*. For each metapath-based rule, $M_i$, in a set of rules, $\mathcal{M} = \{M_1, M_2, ..., M_m\}$, we denote the rule weight by $w(M_i) \in \mathbb{R}$, where $0 \leq w(M_i) \leq 1$. Such a weight indicates the relative usefulness of the metapath-based rule to the prediction task. In Section 3.3, we discuss how we initialize and compute these weights.

## 3.2 OVERVIEW OF MARS.

Using a deep RL process, MARS trains an agent to take walks of length $L$ through the KG to connect pairs of nodes having the pre-defined relation of interest. Here, that relation is $\texttt{induces}(Drug, BP)$, which are masked from the agent during training. Each walk generates a path, $P$, such as the one in the previous section 3.1. This path, $P$, can also be understood as a series of $L$ transitions: $P := (e_c \xrightarrow{r_1} e_2 \xrightarrow{r_2} ... \xrightarrow{r_L} e_{L+1})$. The agent may also remain at its current node. Ultimately, the goal of the agent is episodic: to find paths in which the starting node, $e_c$ (the drug), and the terminal node, $e_{L+1}$ (the BP) have the $\texttt{induces}$ relation. By training the agent to do so, it can identify node pairs with the desired relationship, thus generalizing beyond the training set to predict novel pairs in a holdout, test set. In other words, while true positive predictions in the test set serve as validation, false positives are positioned as potentially *novel* $\texttt{induces}(Drug, BP)$ predictions.

Similarly to a Markov Decision Process (Bellman, 1957), the agent moves based on its current position and the next possible actions. Additionally, however, the history of the agent's previous actions are encoded with an LSTM (Hochreiter & Schmidhuber, 1997; Sherstinsky, 2020), whose parameters are trained to optimize the reward function (Eq. 5 in Appendix A.4), which is evaluated each time the agent completes $L$ transitions from some starting node.

In short, the reward function, originally from Liu et al. (2021), quantifies how successful $P$ is according to two rewards (Fig. 2). The first, base reward indicates whether the terminal node in the path, $e_{L+1}$, is one of the desired target (BP) nodes ($e_d$) that forms a true pair with the starting (drug) node, $e_c$. Put simply: "given an $\texttt{induces}(Drug, BP)$ triple, did the agent make a successful traversal between the drug and BP nodes?"

The second, supplementary reward, contingent upon the first, indicates whether the corresponding metapath, $\tilde{P}$, matches any metapath-based rule, $M_i$. The second reward is also proportional to the rule's corresponding weight, and MARS updates these weights during training. We accomplish these updates through a novel algorithm we call *two-hop joint probability*, or $P_{2H}$. Therefore, the agent is not only encouraged to find connections between true pairs of nodes, but it is also guided toward paths which resemble known MoAs. Thus, MARS has two key interpretable features: (1)

paths between nodes which as potential MoA predictions, and (2) learned rule weights which serve as a proxy for the importance of each rule.

### 3.3 MARS DYNAMICALLY UPDATES RULE WEIGHTS

After MARS is executed, its learned rule weights reflect each rule's relative usefulness in the prediction task. Additionally, during training, weight updates drive the agent toward more informative paths and bypass the assumptions that pre-assigned rule weights are correct. This eliminates the need for pre-computed or literature-derived rule weights; thus, we initialize all weights uniformly as 0.5, a *medium* level of importance. We test *two* approaches for updating rule weights.

**Naive updates**. The *naive* way to implement weight updates (MARS$_{naive}$) is to increase weights according to the frequency at which each metapath-based rule is satisfied. We record observed frequency, $O$, at which each metapath-based rule is satisfied. This is normalized by the batch-specific expected frequency, $E$, which assumes that every rule has a uniform probability across the total number of occurrences in that batch. If the agent finds zero occurrences, no weight updates are made. Ultimately, this produces a metric, $\mu$ (Eq. 1) in which $\mu > 1$ indicates usefulness (the agent used that rule more than others), and $\mu < 1$ indicates otherwise.

$$\mu_{M_i} = O_{M_i}/E_{M_i} \tag{1}$$

Notably, the value of $\mu$ is bound to avoid division by zero and extreme values. To adjust weight updates relative to batch size and rollouts, we define the minimum and maximum bounds on $\mu$ in Eqs. 2 and 3, where $\rho$ is the total number of metapath-based rules:

$$\mu_{min} = \frac{\rho}{\text{batch size} \times \text{rollouts}} \tag{2}$$

$$\mu_{max} = \rho \times \text{batch size} \times \text{rollouts} \tag{3}$$

Using Eq. 4, denoted by $\Phi$, $\mu$ (Eq. 1) is used to update the weight of a rule, $w(M_i)$. Eq. 4 is regularized by the hyperparameter $\alpha \in \mathbb{R}$, where $0 \leq \alpha \leq 1$ to control how subtle or drastic the weight update is, respectively. If $\alpha = 0$, no weight updates are made. Therefore, $\alpha$ can be selected based on user needs or via hyperparameter optimization.

$$\Phi(\mu, w(M_i)) = w(M_i) \times 2\alpha(\frac{\mu - 1}{\mu + 1}) \tag{4}$$

**2-hop joint probability $P_{2H}$ updates**. The second, more complex method to update weights is based on a term we coined, *two-hop joint probability*, $P_{2H}$. Pseudocode for $P_{2H}$ can be found in Algorithm 1 below. This metric approximates the usefulness of metapath-based rules based on full *and* partial matches. Since the metapaths constituting rule bodies contain several consecutive triples, each two-hop fragment is extracted as in the following example:

$M_{example} := \mathtt{induces}(A, E) \Leftarrow \mathtt{upregulates}(A, B) \wedge \mathtt{interacts}(B, C) \wedge$
$$\mathtt{interacts}(C, D) \wedge \mathtt{participates}(D, E)$$

where two-hop fragments are pairs of binary predicates which share a variable:

- $Fragment1 : \mathtt{upregulates}(X, Y) \wedge \mathtt{interacts}(Y, Z)$
- $Fragment2 : \mathtt{interacts}(X, Y) \wedge \mathtt{interacts}(Y, Z)$
- $Fragment3 : \mathtt{interacts}(X, Y) \wedge \mathtt{participates}(Y, Z)$

Here, the probability of each metapath-based rule is computed as the joint probability of its fragments. For example, the $P_{2H}$ metric for the above metapath-based rule would be computed as $P_{2H}(M_{example}) = p(Fragment1) \times p(Fragment2) \times p(Fragment3)$. We note two caveats. Firstly, to account for partial metapath matches, we relax our definition of conjunction here, allowing truth to be evaluated on the fragment level. Secondly, metapath fragments do not necessarily

represent independent events. To avoid complex computation involving conditional probabilities (Russell & Norvig, 2010), we assume independence, and the $P_{2H}$ metric serves as an *approximation* for the empirical probabilities of metapath-based rules.

Ultimately, MARS with $P_{2H}$ updates (MARS$_{P_{2H}}$) uses *all* information from successful trajectories. Compared to naive updates, $P_{2H}$ updates differ in that (1) Eq. 1 is computed based on the observed and expected probabilities of two-hop fragments, rather than whole metapaths, and (2) $\rho$ within Equations 2 and 3 is the number of unique two-hop fragments possible.

---

**Algorithm 1** $P_{2H}$ weight updates

---

**for** each batch, $\beta$ **do**
    $\mathcal{F} \leftarrow$ [empty list]
    **for** each path, $\mathcal{P}$, that the agent traverses, **do**
        **if** the agent found a true pair **then**
            $\hat{P} \leftarrow$ metapath($\mathcal{P}$)             ▷ extract the metapath
            $\mathcal{F} \leftarrow \mathcal{F} +$ extract_fragments($\hat{P}$)     ▷ a list of two-hop fragments seen
        **end if**
    **end for**
    $E \leftarrow 1/$ num. unique fragments in $\mathcal{F}$
    **for** each unique fragment, $f$, in $\mathcal{F}$ **do**
        $O_f \leftarrow$ count($f$)
    **end for**
    **for** each metapath-based rule body, $M_i$, in $\mathcal{M}$ **do**
        $\theta \leftarrow$ extract_fragments($M_i$)         ▷ a list of the fragments in the metapath
        $P_{2H}(M_i) \leftarrow \prod_{f=1}^{len(\theta)} \frac{O_f}{E}$     ▷ ratio of observed / expected frequency, as in Eq. 1
        $w(M_i) \leftarrow \Phi(P_{2H}(M_i), w(M_i))$     ▷ use Eq. 4 to adjust rule weight
    **end for**
**end for**

---

Implementation details and hyperparameter selection are described in Appendices A.2 and A.6, respectively.

## 3.4 DATASETS: *MoA-net* AND ITS VARIANTS

We design our KG, *MoA-net*, specifically for MoA prediction. *MoA-net* consists of drugs, proteins, and BPs (Appendix A.5). We assemble it using the causal relations between drugs and proteins ==from several real-world datasets comprising experimental data, including== Custom KG (Rivas-Barragan et al., 2020) and OpenBioLink KG (Breit et al., 2020). The BP nodes come from ==experimentally-derived and expert-curated== molecular function annotations in UniProt (Consortium, 2015).

To predict drug-BP triples, which are unique to *MoA-net*, we make use of publicly available functional and biochemical assays in ChEMBL (v33), an open access database of bioactive compounds (Gaulton et al., 2012). Of the 1,622 drug-BP triples obtained, 48 also had *known* MoAs in *Drug-MechDB* (Gonzalez-Cavazos et al., 2023), a manually-curated compendium of MoAs. Between the three node types, we define five unique edge types, or relations, shown in Appendix A.5. We also include all inverse relations, running in the opposite direction of causality.

Using the *hetnetpy* package (Himmelstein et al., 2021), we extract all metapaths (see Section 3.1) from *MoA-net* which we considered to be valid MoAs: those comprising directed, *mechanistic* paths between drug and BP nodes (see Appendix A.1). ==We exclude metapaths depicting *associative* patterns, such as those leveraging information about shared BP targets, from our set of metapath-based rules.== Based on MoAs found in *DrugMechDB*, we limit metapaths to a maximum length of four relations (or *hops*).

Finally, we create variants of *MoA-net*. To investigate reasoning shortcuts, we use the Zietz et al. (2024) implementation of XSwap (Hanhijärvi et al., 2009), which swaps edges in a KG without affecting the distribution of node degrees. We call the resultant KG *MoA-net-permuted*. Additionally, we implement an automatic trimming step, which reduces edges of each class to below a user-specified threshold by iteratively removing those between the highest-degree nodes. By setting the

threshold to 10,000 (thereby reducing protein-protein interactions to $\sim 50\%$ of edges), our approach can work on a subgraph of the *MoA-net*, which we refer to as *MoA-net-10k*.

### 3.5 EVALUATION

We split the drug-BP triples within *MoA-net* into training (60%), validation (20%), and test (20%) sets. We evaluate the models using Hits@$k$, where $k \in \{1, 3, 10\}$ and mean reciprocal rank (MRR), optimizing for the latter. Hits@$k$ reports the proportion of times the correct results are in the top $k$ ranked entries, while MRR reports how highly ranked the first correct item is amongst ranked results (Chen et al., 2020). In addition to these *standard* metrics, we report the *pruned* metrics: these are computed on a subset of the predictions that utilized one of the pre-defined metapath-based rules (see Appendix A.1), excluding all predictions which did not satisfy a rule. Notably, pruned metrics help us assess the *calibration* desideratum, as introduced in Section 1, since rule-based predictions follow the expected model semantics.

We conduct an extensive benchmark of our method against nine different baseline KG embedding (KGE) models, two state-of-the-art NeSy methods, and one network measure (Appendix **??**). We train and evaluate these models on the same data splits as MARS on *MoA-net-10k*.

Finally, MARS$_{P_{2H}}$ has two key interpretable features: firstly, all successful trajectories are recorded, serving as potential MoA predictions. This allows us to compare the predicted MoAs of 48 drug-BP pairs against their known MoAs (see Section 3.4). Secondly, learned rule weights serve as a proxy for the importance of each metapath-based rule. This helps determine whether agent trajectories are biased toward certain types of paths. Alongside the pruned metrics, these features help evaluate MARS' alignment with domain knowledge.

## 4 RESULTS

### 4.1 ASSOCIATIVE PATTERNS IMPROVE ACCURACY BUT OFFER LIMITED PRACTICAL USE

In an initial set of experiments on *MoA-net*, we observed that pruned metrics were consistently lower than standard ones (Fig. 3-A), indicating that the metapath-based rules were not being utilized in most predictions. This can happen because rule-based rewards are contingent upon a true pair being found (Eq. 5). Additionally, amongst recorded trajectories, most did *not* follow our metapath-based rules; instead, most trajectories used the following *associative* pattern, involving inverse edges:

$$\texttt{induces}(Drug_1, BP_2) \Leftarrow \texttt{induces}(Drug_1, BP_1) \wedge$$
$$\texttt{induces}(Drug_2, BP_1) \wedge \texttt{induces}(Drug_2, BP_2)$$

This associative pattern indicates that two drugs inducing a common BP also likely induce another BP. This type of pattern is also present in Liu et al. (2021), in which the most used pattern was the following:

$$\texttt{treats}(Drug_1, Disease) \Leftarrow \texttt{causes}(Drug_1, Side\ Effect) \wedge$$
$$\texttt{causes}(Drug_2, Side\ Effect) \wedge \texttt{treats}(Drug_2, Disease)$$

However, when we reproduced the results from Liu et al. (2021), we achieved the same reported metrics even in the absence of the above, associative rule (See Appendix A.8). This suggests that, although the associative rule may serve as a *plausible* model explanation, it does not necessarily guide model training. Furthermore, as stated in Section 1, MoAs, like those in *DrugMechDB*, involve physical, molecular interactions, rather than associative ones.

### 4.2 $P_{2H}$ UPDATES REVEAL REASONING SHORTCUTS VIA DEGREE BIAS

In addition to analyzing the agent trajectories, we used $P_{2H}$ to assess how informative each of the metapath-based rules is in making predictions. In particular, MARS$_{P_{2H}}$ weights showed that paths involving consecutive protein-protein interactions (PPIs) (*i.e.,* $\texttt{interacts}(Protein, Protein)$), were consistently less important (Fig. 4). This indicated that the agent avoided exploring consecutive PPIs.

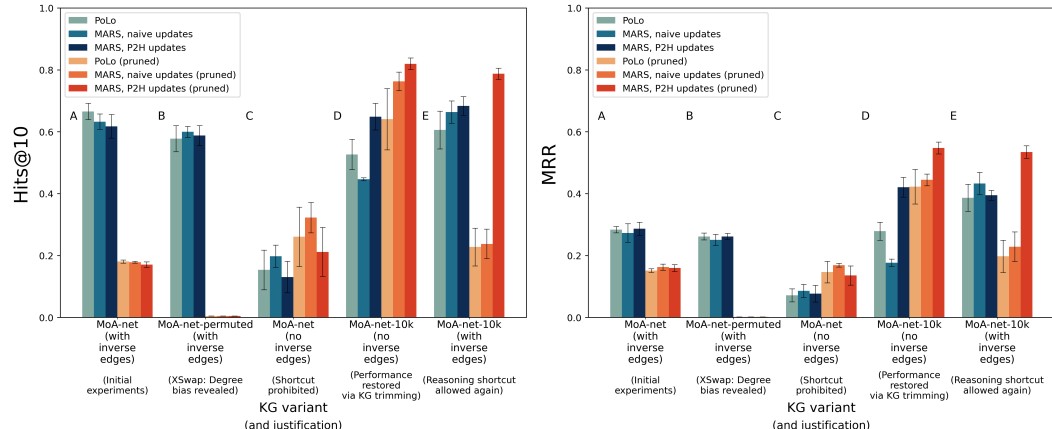

Figure 3: Hits@10 and MRR for MARS$_{P_{2H}}$ compared to PoLo and MARS$_{\text{naive}}$ upon several variants of *MoA-net*. Each bar is the average and standard deviation across five independent training and testing iterations. From left to right: Little change between initial metrics upon *MoA-net* (A) in comparison to the standard *MoA-net-permuted* metrics (B) provides evidence that predictions are influenced by degree bias, resulting in a reasoning shortcut. Thereafter, inverse edges were removed to prohibit the reasoning shortcut, hindering performance (C). Performance was restored upon MoA-net-10k with the KG trimming step (D), with MARS$_{P_{2H}}$ showing the best standard and pruned metrics. Finally, MARS$_{P_{2H}}$ maintains high pruned metrics even when inverse edges (and reasoning shortcuts) are re-introduced (E).

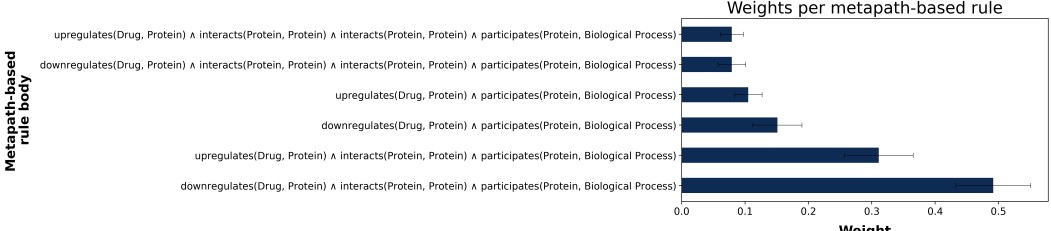

Figure 4: Metapath-based rule weights from MARS$_{P_{2H}}$ on *MoA-net* (Fig. 3-A). Each bar is the average and standard error across five independent training and testing iterations. Paths involving consecutive PPIs (`interacts`($Protein, Protein$)), the most common relation type, have consistently lower weights.

Previous research on KGs has shown that node degree distribution, the number of adjacent edges for each KG node, can significantly bias predictions (Tang et al., 2020; Ju et al., 2024). Specifically, *inspection bias*, a type of degree bias, occurs when the KG is not uniformly inspected or sampled (Zietz et al., 2024). Since PPIs are the most common relation type in *MoA-net* (90% of edges) (Appendix A.5), protein nodes have a higher degree distribution than other node types. We hypothesized, therefore, that the agent circumvents denser parts of the KG, creating an inspection bias. Although rule-based predictions merit a larger reward, the MARS agent exploits associative patterns for a more reliable reward. To confirm the existence of degree bias, we tested our approach upon *MoA-net-permuted*. As explained in Section 3.4, *MoA-net-permuted* is a variant of *MoA-net* in which edges are swapped while preserving node degree distribution. This tests the extent to which node degree drives predictions. Indeed, the lack of change amongst standard performance metrics suggested that node degree was largely responsible for predictions (Fig. 3-B). Put simply, the agent gets lost when exploring the PPIs, so it avoids them.

### 4.3 IDENTIFYING AND MITIGATING DEGREE BIAS IMPROVES PERFORMANCE

To temporarily prohibit the models from using associative patterns as in Section 4.1, we removed inverse edges from *MoA-net* and corresponding metapath-based rules. Consequently, the performance metrics were poor, (*e.g.,* MRR consistently $< 0.1$ (Fig. 3-C)). This confirmed that the models relied on associative patterns for predictions.

Next, we wanted to confirm that the agent was getting lost within the PPIs. As explained in Section 3.4, *MoA-net-10k* is a variant of *MoA-net* with fewer PPIs. We tested MARS$_{P_{2H}}$, MARS$_{\text{naive}}$, and PoLo with the same parameters upon on *MoA-net-10k* (Fig. 3-D). As before, we excluded inverse edges. Since we set trajectory length $L = 4$, our approach automatically removed drug-BP triples from the validation/test sets that were no longer connected via directed paths of length $\leq 4$, resulting in 100 and 90 triples, respectively. Metrics were markedly improved for PoLo, MARS$_{\text{naive}}$, and *particularly* for MARS$_{P_{2H}}$, in comparison to the full *MoA-net* without inverse edges (Fig. 3-C). To ensure this improvement was not simply the result of a reduced test set, we also tested the approaches upon *MoA-net* with 100 sampled test triples, which showed no change (see Appendix A.9).

While removing inverse edges improved metrics, a shortcut-aware system should achieve high *performance* even with the shortcut present (Marconato et al., 2024a). We addressed this next.

### 4.4 MARS$_{P_{2H}}$ RETAINS PERFORMANCE AMONGST RULE-BASED PREDICTIONS

We re-introduced inverse edges to *MoA-net-10k*, thereby restoring the ability to use reasoning shortcuts. Thereafter, we tested each of MARS$_{P_{2H}}$, MARS$_{\text{naive}}$, and PoLo again (Fig. 3-E). While each approach was optimized for standard MRR, pruned metrics indicated how well positive predictions aligned with rules. In Fig. 3-E, we show that both MARS variants and PoLo achieved standard metrics on par with or better than *MoA-net-10k* without inverse edges (Fig. 3-D). However, MARS$_{P_{2H}}$ also achieved pruned metrics comparable to its standard metrics, showing improved *calibration* relative to PoLo and MARS$_{\text{naive}}$. Finally, as in section 4.2, we used XSwap on *MoA-net-10k* to assess the susceptibility of MARS$_{P_{2H}}$ to degree bias. Unlike in Section 4.2, we found no evidence for degree bias (see Appendix A.10).

### 4.5 EXTERNAL VALIDATION OF MARS$_{P_{2H}}$ ON *MoA-net-10k*

In comparison to baseline methods, MARS$_{P_{2H}}$'s metrics outperformed all but MINERVA's, with which they were comparable (Table 1). However, since MINERVA does not, by design, utilize rules for guidance, it suffers the same reasoning shortcuts as PoLo and MARS$_{naive}$. In contrast to MINERVA, DWPC suffers the opposite limitation: predictions are based *only* on metapath-based rules. MARS$_{P_{2H}}$'s *pruned* metrics, which are directly comparable, also outperform DWPC. Finally, as mentioned in Section 3.4, several drug-BP pairs corresponding to known MoAs in *DrugMechDB* were included in the *MoA-net* test set. Of these, 33 pairs remained within *MoA-net-10k*'s test set, and MARS$_{P_{2H}}$ recovered the correct MoA for all of them. Thus, this comprehensive benchmark highlights MARS' ability to achieve near state-of-the-art performance by effectively balancing domain-specific knowledge with the capacity to generalize beyond it.

## 5 DISCUSSION

NeSy approaches are sometimes portrayed as more trustworthy than their black-box counterparts, partially due to increased interpretability (Gaur & Sheth, 2024; DeLong et al., 2024). Here, we presented a NeSy RL approach, MARS$_{P_{2H}}$, which promotes interpretability by deconvoluting drug MoAs. Specifically, through our novel algorithm, two-hop joint probabilities ($P_{2H}$), MARS learned weights corresponding to rules representing MoA patterns; each weight served as a proxy for each rule's importance. However, these insights revealed a new issue: NeSy RL approaches on KGs are susceptible to reasoning shortcuts. Specifically, in our study, predictions were driven by node degree bias. Ultimately, MARS' interpretability called the trustworthiness of such approaches to question.

To address this, we considered Marconato et al. (2024a)'s desiderata for a shortcut-aware NeSy system. Specifically, on *MoA-net-10k*, MARS$_{P_{2H}}$ showed both competitive *performance* as well as *calibration* in comparison to other models. Notably, however, measuring *calibration* is challenging

Table 1: Performance of MARS upon *MoA-net-10k* against baseline models. Metrics are presented as (*average*, *standard deviation*) across five independent training/testing iterations for all but DWPC, which is deterministic. The best of each standard (top) and pruned (bottom) metric are in bold.

| Model | metric type | Hits@1 | Hits@3 | Hits@10 | MRR |
|---|---|---|---|---|---|
| CompGCN | standard | (0.093, 0.010) | (0.212, 0.031) | (0.428, 0.043) | (0.201, 0.011) |
| ComplEx | standard | (0.141, 0.025) | (0.287, 0.020) | (0.517, 0.007) | (0.258, 0.018) |
| MuRE | standard | (0.066, 0.020) | (0.157, 0.043) | (0.377, 0.025) | (0.160, 0.023) |
| PairRE | standard | (0.131, 0.023) | (0.296, 0.035) | (0.601, 0.028) | (0.271, 0.022) |
| RotatE | standard | (0.132, 0.040) | (0.190, 0.042) | (0.349, 0.022) | (0.198, 0.031) |
| MINERVA | standard | **(0.342, 0.016)** | **(0.516, 0.042)** | (0.66, 0.066) | **(0.45, 0.026)** |
| PoLo | standard | (0.272, 0.041) | (0.462, 0.054) | (0.606, 0.061) | (0.387, 0.044) |
| MARS$_{naive}$ | standard | (0.33, 0.031) | (0.482, 0.066) | (0.664, 0.036) | (0.433, 0.036) |
| MARS$_{P_{2H}}$ | standard | (0.23, 0.007) | (0.492, 0.027) | **(0.684, 0.03)** | (0.395, 0.016) |
| Metapaths with DWPC | pruned | 0.370 | 0.560 | 0.780 | 0.508 |
| PoLo | pruned | (0.17, 0.049) | (0.228, 0.061) | (0.228, 0.061) | (0.198, 0.052) |
| MARS$_{naive}$ | pruned | (0.22, 0.049) | (0.238, 0.048) | (0.238, 0.048) | (0.229, 0.048) |
| MARS$_{P_{2H}}$ | pruned | **(0.394, 0.026)** | **(0.644, 0.034)** | **(0.788, 0.018)** | **(0.535, 0.02)** |

in this domain. While rule-based predictions, measured through pruned metrics, follow the expected semantics for MoA deconvolution, we can not determine whether every *other* prediction follows *unintended* semantics. For example, in the classic MNIST addition task, popularly used to assess NeSy methods (Manhaeve et al., 2018), a model is trained to determine the sum of two handwritten digits. In this toy example, the misclassification of a handwritten '2' as '3' and vice versa would still amount to the same sum. Thus, reasoning shortcuts can be objectively identified. On the contrary, while we provide evidence that predictions using associative patterns are *largely* affected by node degree bias, we can not determine whether such patterns *always* reflect a reasoning shortcut.

Finally, regarding *cost effectiveness*, MARS$_{P_{2H}}$ can be applied to any KG, serving as a generalizeable mitigation strategy. However, we also note that this was achieved upon *MoA-net-10k*, a trimmed version of *MoA-net*. While we automated this trimming step, such a strategy does not make use of all available information. To scale MARS$_{P_{2H}}$ to denser KGs and maintain its shortcut-aware status, several future directions could be explored. For instance, one could merge similar, high-degree nodes or rely upon domain knowledge, like the identification of promiscuous proteins (Copley, 2020), to make more informed choices about edge trimming or masking. In addition to addressing these methodological limitations, prospective studies could explore more complex MoAs, include binding or expression values, or involve a variety of protein subclasses.

In summary, our study highlights a key concern in which the behavior of some NeSy RL approaches could be attributed to node degree bias, rather than meaningful, domain-specific concepts. The interpretability of our approach, MARS$_{P_{2H}}$, allowed insight into this reasoning shortcut. Therefore, we question whether such shortcuts are identifiable amongst black-box approaches. Additionally, by testing a NeSy approach upon a novel applied task, MoA deconvolution, we could flag down patterns, like associative ones, which were plausible yet arguably less meaningful to biomedical researchers. Therefore, our study emphasizes the importance of testing interpretable models, like NeSy ones, in an applied domain. Finally, while our study honors the desiderata for shortcut-aware NeSy systems, we also examined the extent to which they were applicable to a biomedical domain.

## 6 CONCLUSIONS

We propose a novel prediction task for NeSy approaches on biomedical KGs: mechanism-of-action (MoA) deconvolution. In contrast to previous DD approaches, MoA deconvolution utilizes model interpretability to uncover the molecular mechanisms behind medicinal drugs. We also constructed a publicly available KG, *MoA-net*, for evaluating this task. To predict drug MoAs alongside indications, we designed the MoA Retrieval System (MARS). Relative to previous NeSy approaches, MARS has enhanced interpretability as it dynamically learns weights corresponding to logical rules. We showed that, with respect to the three desiderata for reasoning-aware NeSy systems, MARS has improved *calibration* and *cost effectiveness* compared to its predecessors, thereby enabling the identification *and* mitigation of a reasoning shortcut based on node degree bias.

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

# A Appendix

## A.1 Selected Metapaths

Table A1: Metapaths representing MoAs. Drugs are represented with a $D$, proteins with a $P$, and biological processes with $BP$.

---

$\text{downregulates}(D, P) \rightarrow \text{participates}(P, BP)$
$\text{upregulates}(D, P) \rightarrow \text{participates}(P, BP)$
$\text{downregulates}(D, P) \rightarrow \text{interacts}(P, P) \rightarrow \text{participates}(P, BP)$
$\text{upregulates}(D, P) \rightarrow \text{interacts}(P, P) \rightarrow \text{participates}(P, BP)$
$\text{downregulates}(D, P) \rightarrow \text{interacts}(P, P) \rightarrow \text{interacts}(P, P) \rightarrow \text{participates}(P, BP)$
$\text{upregulates}(D, P) \rightarrow \text{interacts}(P, P) \rightarrow \text{interacts}(P, P) \rightarrow \text{participates}(P, BP)$

---

We use the MoAs in DrugMechDB (Gonzalez-Cavazos et al., 2023) as guidance for the types of MoA patterns which should exist within our selected metapaths. To get MoAs most relevant for our study, we extracted paths between drugs and BPs within DrugMechDB. All of such paths were $\leq 4$ hops long, justifying the maximum length of paths in Table A1:

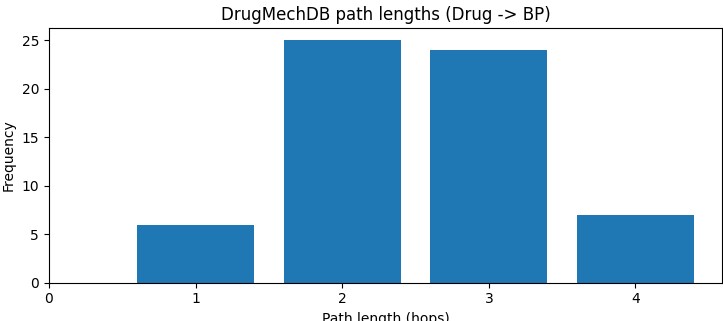

Figure 5: DrugMechDB paths extracted between drugs and BPs.

## A.2 Implementation

We implemented MARS using TensorFlow (version 2.10). The method is packaged in Python and released here [hidden]. The neural network structure is implemented as in Liu et al. (2021), which is also drawn from Das et al. (2018). We used the Adam optimizer (Kingma & Ba, 2015) with REINFORCE (Williams, 1992) to maximize rewards. We used a grid search hyperparameter optimization (Feurer & Hutter, 2019); further details are within Appendix A.6. MARS was trained to optimize MRR, with early stopping determined by validation MRR (Fig. 6).

Partially based on previous biomedical KG benchmarks (Rivas-Barragan et al., 2022), the KGE baseline models include ComplEx (Trouillon et al., 2016), RotatE (Sun et al., 2019), MuRE (Balazevic et al., 2019), CompGCN (Vashishth et al., 2020), and PairRE (Chao et al., 2021). We also compare against PoLo (Liu et al., 2021) and its predecessor MINERVA (Das et al., 2018), which is not guided by rules. Additionally, we test prioritization of drug-BP triples based on degree-weighted path count (DWPC) using 0.4 as damping exponent (Himmelstein & Baranzini, 2015).

The baseline KGE models were trained using the PyKEEN framework (v1.10.1) (Ali et al., 2021). KGEMs were trained using PyKEEN's hyperparameter optimization pipeline over 30 trials using as initial parameters the best configurations from (Rivas-Barragan et al., 2022). The evaluation in the hyperparameter optimization was conducted using Hits@10 for all the models on a link prediction task for the previously-described splits. Network algorithms were implemented in NetworkX (v3.1)

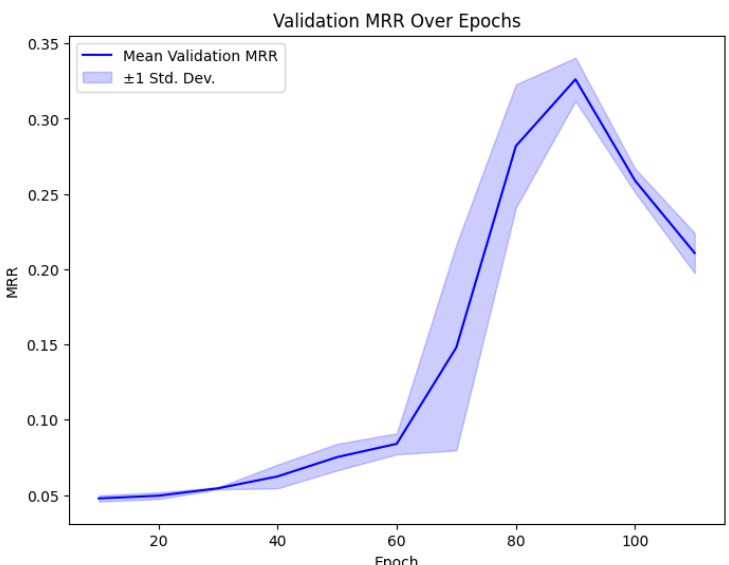

Figure 6: Validation MRR over training epochs.

(Hagberg et al., 2008) and metapaths were calculated using the hetnetpy Python package (Himmelstein et al., 2021; Himmelstein & Baranzini, 2015). Lastly, source code and data are available at [hidden].

### A.3 HARDWARE AND RESOURCES

For training MARS, we used one A40 Nvidia GPU (NVIDIA Corporation, 2021) and two AMD EPYC Milan 7413 CPU nodes (AMD, 2021). On *MoA-net*, MARS used up to 90 Gigabytes of memory and took up to 1.5 hours between beginning training and concluding testing.

### A.4 REWARD FUNCTION

Using a deep RL process, MARS trains an agent to take walks of length $L$ through the KG to connect pairs of nodes having the pre-defined relation of interest (*e.g.*, `induces`); such edges are masked from the agent during training. Each walk generates a path, $P$, which can be understood as a series of $L$ transitions: $P := (e_c \xrightarrow{r_1} e_2 \xrightarrow{r_2} ... \xrightarrow{r_L} e_{L+1})$. The agent may also remain at its current node. Ultimately, the goal of the agent is episodic: to find paths in which the starting node, $e_c$, and the terminal node, $e_{L+1}$, make up one of the true input pairs. By training the agent to do so, it can identify node pairs with some desired relationship, thus generalizing beyond the training set to predict novel pairs. Through a process akin to a Markov Decision Process (Bellman, 1957), the agent makes decisions about its next move based on information about its current position and the next possible actions. Additionally, however, the history of the agent's previous actions are encoded with an LSTM (Hochreiter & Schmidhuber, 1997; Sherstinsky, 2020), whose parameters are trained to optimize the reward function, $R(S_{L+1})$ (Eq. 5), which is evaluated each time the agent completes $L$ transitions from some starting node, reaching a state $S_{L+1}$. MARS uses the same reward function as Liu et al. (2021):

$$R(S_{L+1}) = \mathbb{1}_{\{e_{L+1}=e_d\}} + \mathbb{1}_{\{e_{L+1}=e_d\}}\lambda \sum_{i=1}^{m} w(M_i)\mathbb{1}_{\{\tilde{P}=M_i\}} \tag{5}$$

$$\mathbb{1}_{\{A\}} = \begin{cases} 1 & \text{if } A = \text{true} \\ 0 & \text{if } A = \text{false} \end{cases} \tag{6}$$

This reward function which quantifies how successful $P$ is according to two summands, where the hyperparameter $\lambda$ influences the balance between them. The first summand indicates whether the terminal node in the path, $e_{L+1}$, is one of the desired target (BP) nodes ($e_d$) that forms a true pair with the starting (drug) node, $e_c$. The second summand, contingent upon the first, indicates whether the corresponding metapath, $\tilde{P}$, matches any metapath-based rule, $M_i$. Therefore, the agent is not only encouraged to find connections between true pairs of nodes, but it is also guided toward paths which resemble known MoAs. Of note, the second summand is proportional to some metapath-based rule weight, which is learned by MARS.

## A.5 NODE AND EDGE TYPE DISTRIBUTION

| Node type | Count |
|---|---|
| $Drug$ | 300 |
| $Protein$ | 9,301 |
| $Biological\ Process\ (BP)$ | 86 |

| Edge type | Count |
|---|---|
| `interacts`$(Protein, Protein)$ | 86,786 |
| `participates`$(Protein, BP)$ | 4,325 |
| `downregulates`$(Drug, Protein)$ | 2,205 |
| `upregulates`$(Drug, Protein)$ | 1,631 |
| `induces`$(Drug, BP)$ | 1,622 |

## A.6 HYPERPARAMETER SELECTION

Here, we describe hyperparameter selection. Table A2 describes the hyperparameter search space for optimization, and Table A3 describes the hyperparameters which were fixed for every model. Table A4 describes the best hyperparameters for the final results in Fig. 3-E.

Table A2: Hyperparameter search space for grid search optimization (Feurer & Hutter, 2019)

| Hyperparameter | Description | Search space |
|---|---|---|
| $\lambda$ (Lambda) | ratio at which the second summand, or reward, is applied relative to the first summand, or reward, in the reward function | {5, 8, 10} |
| $\alpha$ (alpha) | how dramatically weight updates should be made (if applicable) | {0.001, 0.01, 0.1} |
| learning rate | learning rate of the optimizer | {0.0001, 0.001, 0.01} |
| hidden size | size of hidden layers | {64, 128, 256} |
| batch size | size of sampled mini-batch for training | {128, 256} |
| rollouts | number of times each query (source-terminal node pair) is made or attempted during training | {50, 100} |
| $\gamma_{baseline}$ (gamma baseline) | discount factor for the baseline as implemented in MINERVA (Das et al., 2018) | {0.05, 0.5} |
| $\beta$ (beta) | entropy regularization factor as implemented in MINERVA (Das et al., 2018) | {0.025, 0.05} |

Table A3: Fixed hyperparameter settings

| Hyperparameter | Description | Value |
|---|---|---|
| embedding size | size of the relation and entity embeddings | 256 |
| LSTM layers | number of LTSM layers | 2 |
| test rollouts | number of times each query (source-terminal node pair) is made or attempted during testing | 50 |
| max branching | maximum number of outgoing edges per node shown to the agent in an episode | 150 |
| $\gamma$ (gamma) | discount factor as implemented in REINFORCE (Williams, 1992) | 1 |
| positive reward | reward for finding a true pair | 1 |
| negative reward | penalty for failing to find a true pair | 0 |

Table A4: Best hyperparameters from Table A2 for the experiments in Fig. 3-E.

| Hyperparameter | MARS$_{P_{2H}}$ | MARS$_{naive}$ | PoLo |
|---|---|---|---|
| $\lambda$ | 10 | 5 | 5 |
| $\alpha$ | 0.001 | 0.001 | - |
| learning rate | 0.0001 | 0.0001 | 0.0001 |
| hidden size | 256 | 256 | 64 |
| batch size | 128 | 256 | 256 |
| rollouts | 100 | 100 | 50 |
| $\gamma_{baseline}$ | 0.05 | 0.5 | 0.5 |
| $\beta$ | 0.025 | 0.05 | 0.05 |

## A.7 ADDITIONAL PERFORMANCE METRICS

Within Fig. 7, we report Hits@1 and Hits@3 as in Fig. 3:

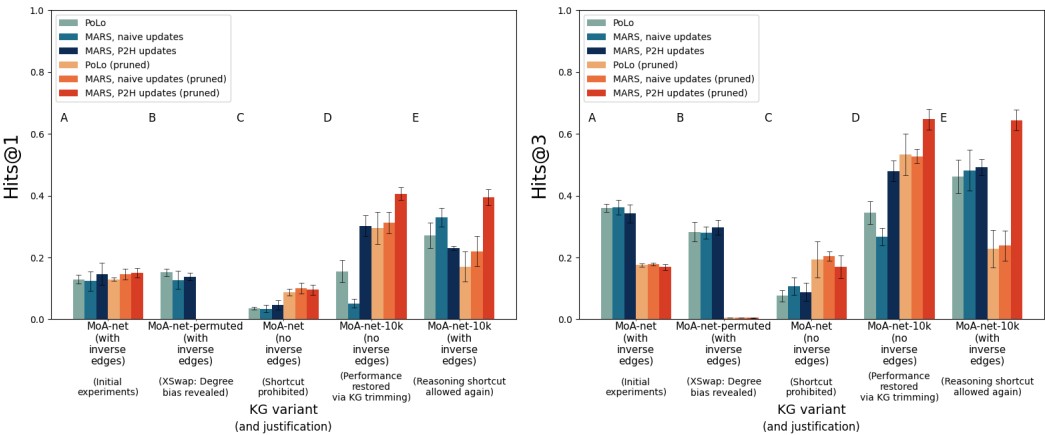

Figure 7: Hits@1 and Hits@3 for MARS$_{P_{2H}}$ compared to PoLo and MARS$_{naive}$ upon several variants of *MoA-net*. Each bar is the average and standard deviation across five independent training and testing iterations. From left to right: Little change between initial metrics upon *MoA-net* (A) in comparison to the standard *MoA-net-permuted* metrics (B) provides evidence that predictions are influenced by degree bias, resulting in a reasoning shortcut. Thereafter, inverse edges were removed to prohibit the reasoning shortcut, hindering performance (C). Performance was restored upon MoA-net-10k with the KG trimming step (D), with MARS$_{P_{2H}}$ showing the best standard and pruned metrics. Finally, MARS$_{P_{2H}}$ maintains high pruned metrics even when inverse edges (and reasoning shortcuts) are re-introduced (E).

## A.8 POLO METRICS WITHOUT ASSOCIATIVE RULES

We ran PoLo on the *Hetionet* KG (Himmelstein et al., 2017) using the same parameters and data splits as reported by Liu et al. (2021). In contrast to Liu et al. (2021), we input all directed metapaths of length $L \leq 4$ as rule bodies (as in Appendix A.1). These metapaths served as the metapath-based rules for PoLo. Notably, these metapaths excluded the associative metapath mentioned in section 4.1:

$$treats(Drug_1, Disease) \Leftarrow causes(Drug_1, Side\ Effect) \land$$
$$causes(Drug_2, Side\ Effect) \land treats(Drug_2, Disease)$$

Despite the most-used metapath-based rule being absent, PoLo achieved the same standard metrics as previously reported (Table A5).

Table A5: Performance evaluations of PoLo upon *Hetionet* as reported in Liu et al. (2021) (*average* across five independent training/testing iterations) and PoLo upon *Hetionet without* associative rules ((*average*, *standard deviation*) across four independent training/testing iterations.)

| rule types | Hits@1 | Hits@3 | Hits@10 | MRR |
|---|---|---|---|---|
| associative ((Liu et al., 2021)) | 0.314 | 0.428 | 0.609 | 0.402 |
| mechanistic (this study) | (0.328, 0.046) | (0.465, 0.037) | (0.656, 0.044) | (0.431, 0.035) |

## A.9 ABLATION STUDY

Here, we tested the effects of reducing the test set size (n=100) on performance. The lack of change between Fig. 8-C and C (test=100) indicates that a reduction in test set size is not responsible for improvements observed in Fig. 8-D.

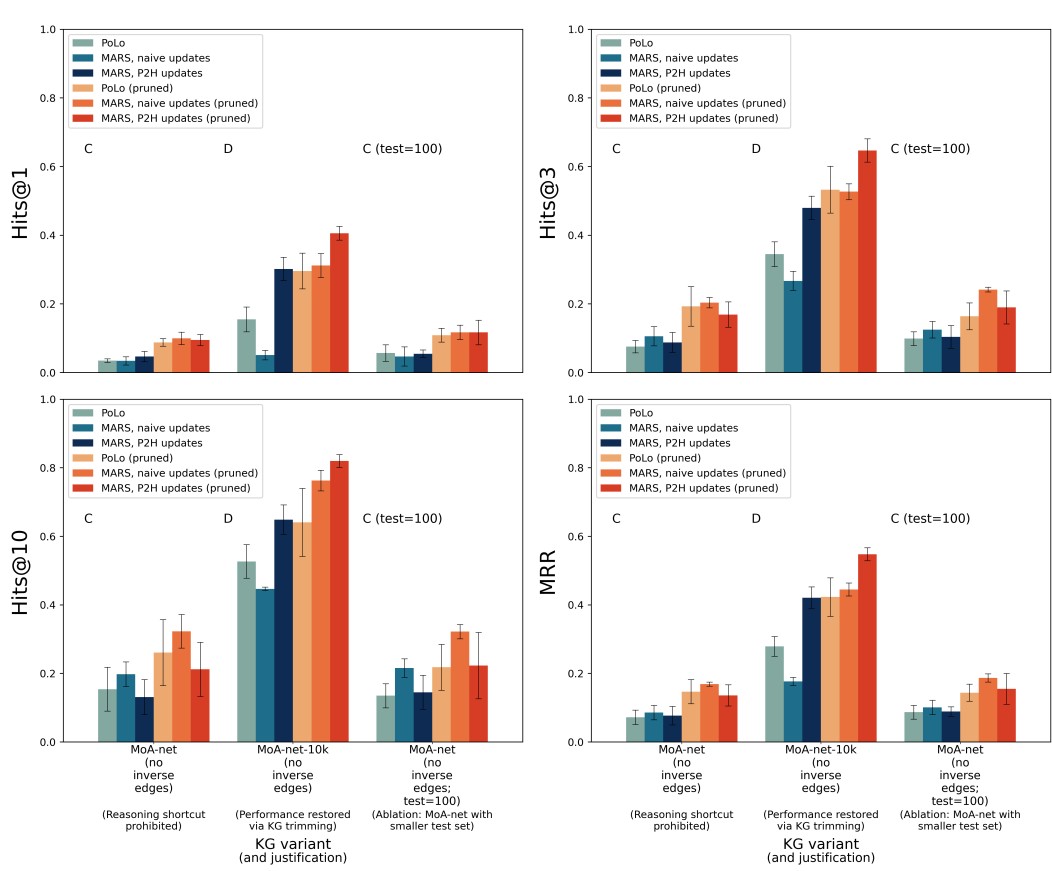

Figure 8: Performance evaluations upon *MoA-net* (no inverse edges) with a test set of 100 triples. Metrics are presented as the average, with error bars representing standard deviation across five independent training/testing iterations. The lack of change between C and C (test=100) indicates that a reduction in test set size is not responsible for improvements observed in D.

### A.10 XSwap permutations: MARS$_{P_2H}$ on *MoA-net-10k*

Using the XSwap algorithm as in Section 4.2, we checked, once again, whether the prediction metrics achieved using MARS$_{P_2H}$ on *MoA-net-10k* were influenced by degree bias. This time, there was a stark decrease in performance metrics upon the permuted KG (Fig. 9). This showed that predictions made by MARS$_{P_2H}$ were due to factors beyond node degree bias.

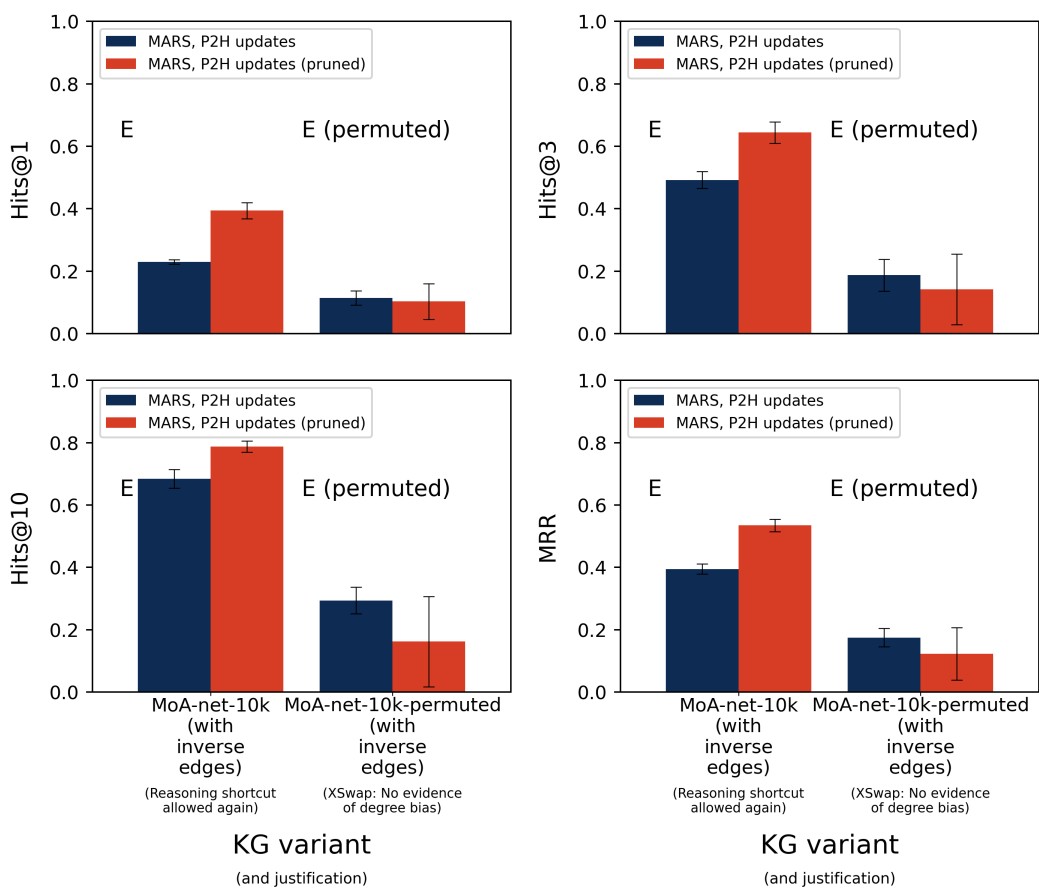

Figure 9: Performance evaluations of MARS$_{P_2H}$ on *MoA-net-10k* as well as a permuted variant of *MoA-net-10k* via the XSwap algorithm. Metrics are presented as the average, with error bars representing standard deviation across five independent training/testing iterations. A drop in performance metrics (E (permuted)) indicates that node degree was not the main driver in predictions made in E.

