# OpenReview forum: "MARS: A neurosymbolic approach for interpretable drug discovery"
_ICLR.cc/2025/Conference — Submitted to ICLR 2025_

### Official Review · Reviewer_FcaQ · 2024-11-01

**Soundness:** 2
**Presentation:** 2
**Contribution:** 2
**Rating:** 5
**Confidence:** 3

**Summary:**

The paper proposes MARS, a neurosymbolic model designed to improve interpretability in drug discovery through a task called Mechanism of Action (MoA) deconvolution, using a specialized knowledge graph named MoA-net. MARS intends to offer insights into the interactions among drug, protein, and biological process via learned rule weights.

**Strengths:**

* The integration of neurosymbolic approaches provides a better understanding of the evidence chain in drug discovery.
* The dynamically learned rule weights may offer an interpretable explanation for the potential impacts of reasoning shortcuts.
* The experimental results are relatively comprehensive and reasonable.

**Weaknesses:**

* The paper is poorly structured and written, and quite difficult to follow.
* Baseline methods compared in the experiments appear to be outdated, with the latest only dating back to 2020.
* Although the authors claim that interpretable symbols can enhance the learning process, there is insufficient experimental support for this assertion, especially regarding the discovery of unseen reasoning paths.
* While this work is presented as a reasonable exploration of biologically meaningful evidence chain reasoning shortcuts, the authors only apply it to three types of entity relationships. Existing study [1] has researched more comprehensive biological pathway evidence chain mining using symbolic reasoning and reinforcement learning. A detailed comparison of the similarities and differences between the two approaches, along with a more substantial discussion of the advantages of your work, is needed.

**References**

[1] Sudhahar, S., Ozer, B., Chang, J. et al. An experimentally validated approach to automated biological evidence generation in drug discovery using knowledge graphs. Nat Commun 15, 5703 (2024).

**Questions:**

* In Lines 235-239, why do the authors state, "Although other metapaths are possible, we exclude them from our set of metapath-based rules"? What's the reason for excluding possible metapaths?
* Following the previous question, what is the rationale for limiting the length of metapaths to 4? Could this lead to insufficient exploration of reasoning paths? If possible, could a comparative experiment on different lengths be conducted?
* In Lines 277-282, the comparison baselines seem to be the latest from 2020. Are there any recent works related to knowledge graphs?
* Why do a significant portion of the results in the experimental tables lack standard deviation? For example, the top 9 baseline models in Table 1.
* Please provide visualizations of the learned rule weights and corresponding analyses.

If the authors could adequately address my concerns, I will consider raising the score.

---

> ### Author Response · Authors · 2024-11-20
> **Response to Reviewer FcaQ's Review**
>
> Thanks to the reviewer for the constructive feedback.
>
> We are glad that the reviewer recognizes the novelty of our study within the implementation of dynamically learned rule weights. We also thank the reviewer for for appreciating the motivation and significance of our work.
>
> Below, we respond to feedback in a point-by-point manner:
>
> - **Reviewer comment:** "The paper is poorly structured and written, and quite difficult to follow."
>
>     **Response:** With the introduction of our new Figures 1 and 2, as well as extensive clarification within the introduction and methodology sections, we believe that our paper is now better-structured and easier to follow.
>
> - **Reviewer comment:** "Baseline methods compared in the experiments appear to be outdated, with the latest only dating back to 2020." + " In Lines 277-282, the comparison baselines seem to be the latest from 2020. Are there any recent works related to knowledge graphs?"
>
>     **Response:** We selected 10 well-established baseline models which have implementations available in the actively maintained and well-regarded PyKEEN library (https://github.com/pykeen/pykeen), so we believe that we have a comprehensive coverage of baseline KG embedding methods.
>
>     Additionally, we made our selection based upon previous work on biomedical KGs. More current methods have not been tested extensively upon biomedical KGs.
>
>     However, per the reviewer's suggestion, we are currently working on implementing one more recent baseline model from the PyKEEN library. We will update our manuscript with the results.
>
> - **Reviewer comment:** "Although the authors claim that interpretable symbols can enhance the learning process, there is insufficient experimental support for this assertion, especially regarding the discovery of unseen reasoning paths."
>
>     **Response:** Assuming that the reviewer is suggesting that the experiments do not support that the inclusion of logical rules improves MARS' predictive capabilities, we refer back to the metrics reported in the new Figure 3, experiment E. This shows that, when we include logical rules, along with weights updated via the P2H approach, MARS is able to circumvent reasoning shortcuts.
>
> - **Reviewer comment:** "While this work is presented as a reasonable exploration of biologically meaningful evidence chain reasoning shortcuts, the authors only apply it to three types of entity relationships. Existing study [1] has researched more comprehensive biological pathway evidence chain mining using symbolic reasoning and reinforcement learning. A detailed comparison of the similarities and differences between the two approaches, along with a more substantial discussion of the advantages of your work, is needed."
>
>     **Response:** Thank you for sharing this study with us. We are currently working on including a comparison to this very recent study within our Related Works section.
>
> - **Reviewer comment:** "In Lines 235-239, why do the authors state, "Although other metapaths are possible, we exclude them from our set of metapath-based rules"? What's the reason for excluding possible metapaths?"
>
>     **Response:** We have now clarified this within Section 3.4:
>
>         "Using the hetnetpy package (Himmelstein et al., 2021), we extract all metapaths (see Section 3.1) from MoA-net which we considered to be valid MoAs: those comprising directed, mechanistic paths between drug and BP nodes (see Appendix A.1). We exclude metapaths depicting associative patterns, such as those leveraging information about shared BP targets, from our set of metapath-based rules. "
>
>     Furthermore, we make the distinction between associative and mechanistic patterns clearer within our new Fig. 1 as well as the third paragraph of introduction:
>
>         "Unfortunately, these associative patterns, which are discovered during or after clinical trials, are rare or absent for novel compounds. Furthermore, such patterns can not represent the MoA of a drug; instead, an MoA involves mechanistic patterns, such as the physical, molecular interactions shown in Fig. 1."

---

> > ### Author Response · Authors · 2024-11-20
> > **Continued Response to Reviewer FcaQ's Review**
> >
> > - **Reviewer comment:** "Following the previous question, what is the rationale for limiting the length of metapaths to 4? Could this lead to insufficient exploration of reasoning paths? If possible, could a comparative experiment on different lengths be conducted?"
> >
> >     **Response:** Thanks for highlighting this. We have revised the sentence in Section 3.4 to be clearer, stating:
> >
> >         "Based on MoAs found in DrugMechDB, we limit metapaths to a maximum length of four relations (or hops)"
> >
> >     We would also like to note that, since the agent has the option, at any point, to remain at its current node, paths shorter than four hops can also be explored. However, all of the MoAs occurring between drugs and BPs in DrugMechDB are four hops long or less.
> >
> >      The suggestion to do a comparative experiment of different lengths is an interesting one. However, since specifying a length of four hops also allows exploration of paths <= 4, we did not feel this was necessary for the goals of our paper.
> >
> > - **Reviewer comment:** "Why do a significant portion of the results in the experimental tables lack standard deviation? For example, the top 9 baseline models in Table 1."
> >
> >     **Response:** Originally, metrics reported for the KG embedding methods were reported based on the best metrics from 30 trials. However, we see that this is inconsistent with how we report MARS. Therefore, we have revised our table to include the average and standard deviation across five trials.
> >
> > - **Reviewer comment:** "Please provide visualizations of the learned rule weights and corresponding analyses."
> >
> >     **Response:** This visualization of learned rule weights is now available in Fig. 4. Metrics for other analyses are in Fig. 3.

---

> > > ### Comment · Reviewer_FcaQ · 2024-11-26
> > >
> > > Thank you for the authors' response and revisions. Some of my concerns have been addressed, and based on this, I have increased my score. However, there are still remaining issues:
> > > 1. **"Baseline methods compared in the experiments appear to be outdated, with the latest only dating back to 2020."**—With just a casual search, I came across updated KGE methods like HousE and RelEns-DSC, and it is highly likely that even more recent algorithms from 2024 are available. It is recommended to select a balanced set of well-established classic algorithms and the latest SOTA methods from the past two years as baselines for comparison, rather than relying heavily on outdated models. Furthermore, while Python libraries are convenient, their development often lags behind the latest research. Over-reliance on simplified interfaces may limit the cutting-edge nature of the comparisons.
> > > 2. **"...Existing study has researched more comprehensive biological pathway evidence chain mining..."**—My primary concern is that, for the same task of mining biological pathway evidence chains, the major findings of this work [3] appear to be more comprehensive and deeper compared to yours. Their approach accommodates a broader range of entity types and is capable of discovering pathways that are currently unknown in the biological domain.
> > >
> > > [1] Rui L., et al. House: Knowledge graph embedding with householder parameterization. ICML 2022.\
> > > [2] Yue L., et al. Relation-aware Ensemble Learning for Knowledge Graph Embedding. EMNLP 2023.\
> > > [3] Sudhahar S., et al. An experimentally validated approach to automated biological evidence generation in drug discovery using knowledge graphs. Nat Commun 15, 5703 (2024).

---

> > > > ### Author Response · Authors · 2024-11-27
> > > > **Additional Response to Reviewer FcaQ**
> > > >
> > > > Dear, reviewer,
> > > >
> > > > Thanks for your commitment to the discussion period and for your constructive feedback. Thanks also for being receptive to the changes we made and altering your score. We would like to respond to your last two concerns below:
> > > >
> > > > 1. As promised in our original response, we implemented two more of the newer baseline models from PyKEEN, CompGCN (2020) and PairRE (2021). We have integrated these results into our manuscript. We have also removed results from some of the older models to save space.
> > > >
> > > >
> > > >     We see, based on your most recent comment, that this may not align with your expectations. However, please bear in mind that, when you had originally expressed the concern, you had implicated a need for newer baselines than the ones we had, without specific years or methods mentioned.
> > > >
> > > >
> > > >     Regarding the two examples you gave, the second one, RelEns, is an ensemble approach, which is an unfair comparison compared to single approaches [1]. We will do our best, over the discussion period, to implement HouseE, as suggested.
> > > >
> > > >
> > > >     Finally, please consider that, even if some methods outperform ours (as MINERVA does), the novelty in our approach also lies in the balance between competitive performance and interpretability, as MARS produces path-like explanations alongside predictions.
> > > >
> > > >
> > > >     [1] Daniel Rivas-Barragan, Daniel Domingo-Fernández, Yojana Gadiya, David Healey, Ensembles of knowledge graph embedding models improve predictions for drug discovery, Briefings in Bioinformatics, Volume 23, Issue 6, November 2022, bbac481, https://doi.org/10.1093/bib/bbac481
> > > >
> > > >
> > > >
> > > > 2. Thanks for pointing this recent study out to us. We see two major differences between that study and ours:
> > > >
> > > >     - **Major Difference 1:** The referenced study uses the output of AnyBURL, a rule-mining method, alongside the output of other KG prediction models. Therefore, since these path-like explanations are derived separately, they are not necessarily faithful to their model's reasoning processes. MARS' rule weights indicate how important each metapath-based rule is, demonstrating how faithful MARS' output paths are to its reasoning processes.
> > > >
> > > >     - **Major Difference 2**. The referenced study has very different goals than us. While both their study and ours seek path-like, biomedical explanations, theirs involves higher-level clinical associations, whereas our study aims to uncover mechanistic, molecular interactions. In addition to the previously mentioned clarifications, we hope that we have made this clearer within Section 2, Related Works:
> > > >
> > > >             "For example, Sudhahar et al. (2024) investigates 'evidence chains', paths explaining associations between drugs and diseases. However, these explanations are derived separately from indication predictions, using an additional rule-mining model (Meilicke et al., 2019) ...
> > > >
> > > >             ... In contrast to these previous studies, we focus upon paths representing MoAs, involving mechanistic, molecular relations."
> > > >
> > > >
> > > > In fact, we believe that the prospective directions listed by the referenced study help to highlight the novelty and usefulness of our work. Specifically:
> > > >
> > > > - They intend to integrate molecular-level information, which is what we included within our KG: "We intend to also integrate data on protein-protein interactions, proteins and biological functions interactions and interactions between biological functions in the knowledge graph to better explain the mechanism of action of the drug in the disease of interest."
> > > >
> > > > - They also state that: "there are no benchmark datasets available to evaluate evidence chains." In contrast, we developed a benchmark task for this (MoA deconvolution, tested upon MoA-net)

---

### Official Review · Reviewer_vFxB · 2024-11-03

**Soundness:** 3
**Presentation:** 2
**Contribution:** 2
**Rating:** 6
**Confidence:** 4

**Summary:**

This study presents the MARS (Mechanism-of-Action Retrieval System), a neurosymbolic (NeSy) approach designed for interpretable drug discovery. MARS aims to improve understanding of drug mechanisms of action (MoAs) using knowledge graphs (KGs) combined with logical rule-based inference. The model introduces a tailored KG called MoA-net, enabling MoA deconvolution. MARS enhances interpretability by assigning weights to logical rules and employing a two-hop joint probability (P2H) metric to improve model calibration and address potential reasoning shortcuts.

**Strengths:**

- MARS applies NeSy methods to drug discovery, a field with a strong demand for interpretability. The study’s introduction of MoA deconvolution as a KG-based task and MoA-net as a novel KG for biomedical applications broadens the use of KGs in drug discovery.
- The authors conduct extensive testing of MARS on both synthetic and real data and provide detailed comparisons to existing NeSy methods, particularly regarding susceptibility to reasoning shortcuts.
- By enhancing interpretability in drug discovery, MARS could help researchers gain insights into drug mechanisms, potentially leading to safer and more effective treatments.

**Weaknesses:**

- The model is primarily tested on synthetic KG data and lacks real-world validation on clinical datasets or pharmacological records, which limits the practical assessment of its interpretability and generalization for drug discovery tasks.
- While the P2H metric is introduced to make MARS shortcut-aware, its reliance on rule weighting and complex computations might hinder scalability, especially in large, densely connected KGs that are typical in biomedical data.
- AMARS mainly addresses computational rather than biological significance.
- MARS is highly susceptible to node degree bias, leading to unintended reasoning shortcuts.

**Questions:**

- Testing the model on real-world datasets, such as drug-protein interactions or clinical outcome data, would strengthen the claims regarding its applicability in drug discovery.
- How does MARS handle drug discovery tasks beyond MoA deconvolution?
- How could biological interpretability be further improved? Considering biological plausibility, could the authors incorporate features like protein binding affinities or pharmacokinetic properties to add biological context to MARS’s interpretability?
- Given the computational cost of two-hop joint probability calculations, what specific modifications or future optimizations are planned to enhance scalability?

---

> ### Author Response · Authors · 2024-11-20
> **Response to Reviewer vFxB's Review**
>
> Thanks to the reviewer for their detailed feedback.
>
> We are pleased that the reviewer understands the novelty of our study, the extent to which we tested MARS, and the greater implications of our study with respect to drug discovery.
>
> Below, we respond to the suggestions made:
>
> - **Reviewer comment:** "The model is primarily tested on synthetic KG data and lacks real-world validation on clinical datasets or pharmacological records, which limits the practical assessment of its interpretability and generalization for drug discovery tasks." + " Testing the model on real-world datasets, such as drug-protein interactions or clinical outcome data, would strengthen the claims regarding its applicability in drug discovery."
>
>     **Response:** On the contrary, our KG is created from real-world, experimental data. We've 	clarified this in Section 3.4, stating:
>
> 	    "We assemble it using the causal relations between drugs and proteins from several real-world datasets comprising experimental data, including..."
>
> 	and:
>
> 	    "The BP nodes come from experimentally-derived and expert-curated molecular function annotations in UniProt"
>
>     Furthermore, we have chosen not to include information from clinical records because we designed a KG exclusively involving data collected from a laboratory, rather than information that is collected during or after clinical trials. As we have now clarified in the introduction and the new Fig. 1, this design facilitates drug discovery for novel compounds, rather than pre-existing drugs, In particular, we state:
>
> 	    "For example, such methods utilize associations regarding a drug’s pharmacological class (Ratajczak et al., 2022), side effects (Liu et al., 2021), or known indications. Unfortunately, these associative patterns, which are discovered during or after clinical trials, are rare or absent for novel compounds. Furthermore, such patterns can not represent the MoA of a drug; instead, an MoA involves mechanistic patterns, such as the physical, molecular interactions shown in Fig. 1."
>
> - **Reviewer comment(s):** "While the P2H metric is introduced to make MARS shortcut-aware, its reliance on rule weighting and complex computations might hinder scalability, especially in large, densely connected KGs that are typical in biomedical data." **+** "MARS is highly susceptible to node degree bias, leading to unintended reasoning shortcuts."
>
>     **Response:** MoA-net has nearly 100 thousand edges, making it a relatively large KG. Both MARS and its predecessor, PoLo, scale to this magnitude (as shown in Fig. 3, result A). This means that MARS does, indeed, scale to larger KGs because it can still reason and make predictions from them in the same way that PoLo and other approaches can. However, we show that, using the P2H updates, MARS' predictions are no longer susceptible to node degree bias, relying instead upon mechanistic patterns resembling MoAs (Fig. 3, result E).
>
> - **Reviewer comment:** "MARS mainly addresses computational rather than biological significance."
>
>     **Response:** Given the scope of the conference as a machine learning venue, we believe that the computational significance of our study is a fitting contribution. Additionally, we are convinced that our study also has biological significance as it addresses an open problem (MoA deconvolution) in the drug discovery domain.
>
> - **Reviewer comment:** "How does MARS handle drug discovery tasks beyond MoA deconvolution?"
>
>     **Response:** MARS predicts (1) biological effects by drugs (drug-BP link prediction) and (2) corresponding mechanistic MoAs (the paths between the drug and BP nodes). MoA deconvolution is a novel prediction task which we created based upon an open problem in the drug discovery domain.
>
>     Testing MARS upon other drug discovery tasks is an interesting prospective direction. In theory, with another KG, MARS could be used to predict side effects, tissue-specific effects, etc. However, these additional tasks were outside the scope of the work.
>
> - **Reviewer comment:** "How could biological interpretability be further improved? Considering biological plausibility, could the authors incorporate features like protein binding affinities or pharmacokinetic properties to add biological context to MARS’s interpretability?"
>
>     **Response:** Since this is one of the first studies applying a neurosymbolic approach on a biomedical task, and we are proposing the novel task of MoA deconvolution, we wanted to keep the study tractable to start. However, the reviewer's suggestions are important prospective directions. We have now included the following in our Discussion section:
>
>         "In addition to addressing these methodological limitations, prospective studies could explore more complex MoAs, include binding or expression values, or involve a variety of protein subclasses".

---

> ### Author Response · Authors · 2024-11-20
> **Continued Response to Reviewer vFxB's Review**
>
> - **Reviewer comment:** "Given the computational cost of two-hop joint probability calculations, what specific modifications or future optimizations are planned to enhance scalability?"
>
>     **Response:** We argue that (1) any increase in computational cost, compared to PoLo, is minimal, given that it only involves multiplication of probabilities, and (2) the additional cost pays off with predictions which are better aligned to domain knowledge.
>
>     As future optimizations, we have included the following within our Discussion section:
>
>         "For instance, one could merge similar, high-degree nodes or rely upon domain knowledge, like the identification of promiscuous proteins (Copley, 2020), to make more informed choices about edge trimming or masking."

---

> > ### Comment · Reviewer_vFxB · 2024-11-27
> >
> > Some comments have been addressed, and I can increase the score of submission to 6.

---

> > > ### Author Response · Authors · 2024-11-27
> > > **Additional Response to Reviewer vFxB**
> > >
> > > Thank you again for your constructive feedback and for considering an update to your official score.
> > >
> > > Please let us know if you have any further questions during the discussion period.

---

### Official Review · Reviewer_hv6o · 2024-11-04

**Soundness:** 3
**Presentation:** 3
**Contribution:** 2
**Rating:** 6
**Confidence:** 3

**Summary:**

The paper proposed a systematic framework for interpretable drug discovery. The model comprises of drug mechanism-of-action (MoA) deconvolution; MoA-net and its variants, a specifically designed knowledge graph based on public biomedical data; MARS, a Neural Symbolic approach with through dynamic rule-weight updates. The work provides enhanced interpretability and strong predictive performances compared to other baselines.

**Strengths:**

The paper is generally well-written and easy to follow. The objectives of the proposed framework clearly outlined and the model effectively handles several specific problems that other comparable methods hold. The proposed method shows competitive performances and provides a reasonable form of interpretability using knowledge graphs.

**Weaknesses:**

- The overview of MARS is unclear, and a figure delineating process would be helpful for reader's understanding. This should include the input-output pairs for the proposed model.
- It is difficult to interpret Figure 3, as the main body does not explain anything about the confidences.
- While the paper proposes the term "drug-discovery", the it unclear in how to find  "novel drugs or chemical compounds". Given that the model is trained on the full data (no test) it could provide some meaningful results.
- The computation cost in training the model could provide valuable insights.
- The term "enhanced interpretability" is difficult to understand. An example for the term (possibly from the data that the authors used) would help better understanding of the term, and the strength of the model.

**Questions:**

- What is the computation time of training the network? How susceptible is the model with inclusion of new data?
- What is the "confidence" in figure 3?
- What is the exact neural network architecture for MARS? What is a simple example of input and output pairs?
- How would we approach in finding drug discovery? Does the model provide any candidate paths or of any sort? If so, how would we approach justification of the result?
- What is the "enhanced interpretability"? Would there be an example of this?

---

> ### Author Response · Authors · 2024-11-20
> **Response to Reviewer hv6o's Review**
>
> Thank you to the reviewer for all of the helpful suggestions.
>
> We are glad that the reviewer recognizes the importance of our paper in highlighting some specific problems that other methodologies may possess. We are also glad that the form of interpretability we're investigating is clear and reasonable, given the application. Additionally, thank you for finding it to be well-written and easy to read.
>
> Below, we respond to the feedback in a point-by-point manner:
>
> - **Reviewer comment:** "The overview of MARS is unclear, and a figure delineating process would be helpful for reader's understanding."
>
> 	**Response:** This is a good suggestion. We created and included an overview figure, which is now Figure 2.
>
> - **Reviewer comment:** " It is difficult to interpret Figure 3, as the main body does not explain anything about the confidences." + " What is the "confidence" in figure 3? "
>
> 	**Response:** Unfortunately, this was a labelling mistake. Previously, instead of "weights," we used the term "confidences." We decided to change the term to "weights" throughout the paper, but overlooked the legend in this figure. This figure, which is now Figure 4, is now updated with "weights" instead of "confidences."
>
> - **Reviewer comment:** "While the paper proposes the term "drug-discovery", the it unclear in how to find "novel drugs or chemical compounds". Given that the model is trained on the full data (no test) it could provide some meaningful results. "
>
> 	**Response:** We split the drug-BP triples in our KG into training, validation, and test sets, as specified in the Evaluation section (section 3.5). However, we realize that this might not be clear throughout the rest of our text, so we've tried to make this and the drug discovery task much clearer through the following changes, all of which are highlighted within the body of the paper:
>
> 	- In Section 3.2, Overview of MARS, we clarify that MARS should predict novel pairs "in the test set."
>
> 	- Directly after, we further clarify: "In other words, while true positive predictions in the test set serve as validation, false positives are positioned as potentially novel ‘induces(Drug, BP)’ predictions."
>
> 	- In the second paragraph of Section 3, we now clarify that our task is a link prediction one.
>
> 	- We've now included a new figure, which is now Figure 1. This, along with the (new) third paragraph  of our introduction, should help clarify why we are focusing upon novel drugs, which are not yet past clinical trials.
>
> - **Reviewer comment:** "How would we approach in finding drug discovery? Does the model provide any candidate paths or of any sort? If so, how would we approach justification of the result? "
>
> 	**Response:** We hope that the above point clarifies the first question here. Regarding the second question, we have now clarified within our new Fig. 2 as well as Section 3.2, that MARS' traversals produce paths as output:
>
> 	"Each walk generates a path, P , such as the one in the previous section 3.1."
>
> 	Finally, regarding the third question, we obtained known MoAs from the carefully curated database, DrugMechDB. We use these known MoAs for comparison against some of the paths which MARS discovers. We state this, explicitly, within Section 3.5:
>
> 		"Finally, MARS-P2H has two key interpretable features: firstly, all successful trajectories are recorded, serving as potential MoA predictions. This allows us to compare the predicted MoAs of 48 drug-BP pairs against their known MoAs."
>
> - **Reviewer comment:** "What is the exact neural network architecture for MARS? "
>
> 	**Response:** The neural network structure is the same as in PoLo, which borrows its structure from MINERVA. We clarify this in our supplementary section A.2, "Implementation".
>
> - **Reviewer comment:** "What is a simple example of input and output pairs? "
>
> 	**Response:** Since it is a link prediction task, the input for training would be known drug-BP pairs. True positive output predictions would involve predicting those same pairs as true. False positive predictions may pose as potentially novel predictions. We now clarify this within our new Fig. 2 as well as the second paragraph of Section 3:
>
> 		"Specifically, we accomplish this through a link prediction task, in which we predict whether edges of type induces exist between Drug and BP nodes. Thereafter, new predictions regarding the induces relation serve as potential therapeutic outcomes for the chemical compound represented by the Drug node. "
>
> - **Reviewer comment:** " The computation cost in training the model could provide valuable insights. " + " What is the computation time of training the network? How susceptible is the model with inclusion of new data?"
>
> 	**Response:** We have now included more details about training, including the time and memory requirements within Supplementary Sections A.2, "Implementation" and A.3, "Hardware and Resources".

---

> ### Author Response · Authors · 2024-11-20
> **Continued Response to Reviewer hv6o's Review**
>
> - **Reviewer comment:** "The term "enhanced interpretability" is difficult to understand. An example for the term (possibly from the data that the authors used) would help better understanding of the term, and the strength of the model. " **+** " What is the "enhanced interpretability"? Would there be an example of this? "
>
>     **Response:** We have made the following changes to clarify:
>
>     - In the third paragraph of the Introduction, we now explain that interpretability is broadly defined. We then clarify the type of interpretability we look for:
>
>                 "Therefore, within this study, we focus upon model interpretability which provides mechanistic insight into drug MoAs".
>
>     - Additionally, our new Figure 1 gives an example as to what an MoA looks like, providing a visual depiction of the above.
>
>     - Finally, by "enhanced interpretability," we refer to the dynamic rule weights which are learned by MARS. This is novel compared to PoLo. We clarify this within the fourth paragraph of the introduction:
>
>                 "unlike its predecessors, MARS achieves enhanced interpretability by learning weights associated with logical rules which resemble MoAs."

---

> ### Comment · Reviewer_hv6o · 2024-11-29
>
> Thank to the authors for making detailed revision on the manuscript.
> The authors have addressed the points that were made and changed the score accordingly.

---

### Official Review · Reviewer_eFxS · 2024-11-04

**Soundness:** 3
**Presentation:** 3
**Contribution:** 2
**Rating:** 5
**Confidence:** 5

**Summary:**

*November 26: after first response, the "Presentation" score has been raised from 2 to 3*

This work proposes a new neurosymbolic (NeSy) model for drug discovery as well as an associated dataset, where the problem is cast as a prediction task for drug-biological process (BP) pairs. More specifically, biochemical knowledge is arranged into a knowledge graph (KG) where entities (vertices) such as drugs, proteins and BP are connected with each other via directed edges. By using a new update rule of edge weights, the new model was able to avoid reasoning shortcuts inherent in previous methods.

**Strengths:**

- Using the $ P_{2H} $ update rule, the new method has been shown, via a series of ablation studies, to avoid reasoning shortcuts due to degree bias.

**Weaknesses:**

- The novelty of the manuscript appears to be limited:
    - The new dataset has fewer kinds of entities than PoLo, and in particular does not appears to include those relevant for therapeutics such as "disease" or "side effect".
    - Similarly, one would expect more kinds of entities beyond a generic "protein" (e.g. receptor, transcription factor, enzyme etc.).
    - Another issue is that all metapaths are chains, like $ \mathrm{interacts}(P_{k}, P_{k + 1}) $, whereas in real-world applications there are many instance of multi-protein complexes for biological functions.
    - The model reward function is almost the same as PoLo (Equation 4): $$ R(S_{L + 1}) = 1_{ e_{L + 1} = e_d  }  + b \lambda  \sum\limits_{i = 1}^{m} s(M_i) 1_{\tilde{P} = M_i} $$ In fact, that paper states that the hyparameter $ b $ can be "set to $ 1_{e_{L + 1} = e_d} $" (p. 382).
- The presentation of the manuscript is not as efficacious as one would hope:
    - Key concepts such as "deconvolution" or $ \land $ (presumably conjunction?) were not defined.
    - The 2-hop probability update ($ \S $A.4 and Alg. 1), which is the major novelty in the manuscript, should be in the main text.
    - The main predictive task was not explicitly formulated: from Figure 1A it appears to be link prediction.
    - Moreover, it would be useful to have the catalogue of different entities--entities interaction types (Figure 1B) in a table.
    - The background colour of Figure 1B makes it harder to read, plus the font sizes for the labels are a bit too small.
    - The citation for PoLo should use the published version (https://link.springer.com/chapter/10.1007/978-3-030-77385-4_22)

**Questions:**

- Could you include more training details e.g. hardware, training time, validation loss plots etc.?
- On what basis were the "feasible" MoA's chosen in Table A1?
- When comparing with PoLo, which rewards function(s) were used?

---

> ### Author Response · Authors · 2024-11-20
> **Response to Reviewer eFxS's Review**
>
> Thank you for taking the time to read and understand our paper. We're glad that the reviewer can see the usefulness of our P2H method.
>
> Regarding the novel aspects of our study, we would like to re-emphasize the novelty of our methodology (P2H updates), the novelty of our prediction task (MoA deconvolution), the novelty of our KG (MoA-net), which is the first biomedical KG to comprise drug-BP triples, and the novelty of our findings (strong evidence that neurosymbolic approaches on KGs are susceptible to reasoning shortcuts).
>
> **With regard to the specific points you have made about novelty:**
>
> - **Reviewer comment:** "The new dataset has fewer kinds of entities than PoLo, and in particular does not appears to include those relevant for therapeutics such as "disease" or "side effect"."
>
> 	**Response:** We have chosen not to include entities like "disease" and "side effect" because we designed a KG exclusively involving data collected from a laboratory, rather than information that is collected during or after clinical trials. As we have now clarified in the introduction and the new Fig. 1, this design facilitates drug discovery for novel compounds, rather than pre-existing drugs, In particular, we state:
>
> 		"For example, such methods utilize associations regarding a drug’s pharmacological class (Ratajczak et al., 2022), side effects (Liu et al., 2021), or known indications. Unfortunately, these associative patterns, which are discovered during or after clinical trials, are rare or absent for novel compounds. Furthermore, such patterns can not represent the MoA of a drug; instead, an MoA involves mechanistic patterns, such as the physical, molecular interactions shown in Fig. 1."
>
> - **Reviewer comment:** " Similarly, one would expect more kinds of entities beyond a generic "protein" (e.g. receptor, transcription factor, enzyme etc.). "
>
> 	**Response:** Since this is one of the first studies applying a neurosymbolic approach on a biomedical task, and we are proposing the novel task of MoA deconvolution, we wanted to keep the KG types simple to start. However, as you suggest, this would be an important prospective direction, for which some preliminary experiments are planned. We have now included the following in our Discussion section:
>
> 		"In addition to addressing these methodological limitations, prospective studies could explore more complex MoAs, include binding or expression values, or involve a variety of protein subclasses"
>
> - **Reviewer comment:** "Another issue is that all metapaths are chains... ... whereas in real-world applications there are many instance of multi-protein complexes for biological functions. "
>
> 	**Response:** This is a good point. However, the validation MoAs we could map from DrugMechDB were all chained. We've added this as a prospective direction, given that data availability for curated MoAs is likely to increase:
>
> 		"In addition to addressing these methodological limitations, prospective studies could explore more complex MoAs, include binding or expression values, or involve a variety of protein subclasses "
>
> - **Reviewer comment(s):** "The model reward function is almost the same as PoLo " **+** " When comparing with PoLo, which rewards function(s) were used? "
>
> 	**Response:** While we acknowledge and credit the PoLo paper for the reward function, we agree that the novelty of MARS lies in the P2H updates rather than the reward function. Therefore, following your next suggestion, we (1) moved the information about 2-hop joint probabilities into the main text, and (2) emphasized the reward function less, including only the necessary details for understanding MARS in the main text.
>
> - **Reviewer comment(s):** "The 2-hop probability update (§A.4 and Alg. 1), which is the major novelty in the manuscript, should be in the main text. "
>
> 	**Response:** We agree with your point; per your suggestion, we have now moved this information and integrated it into the main text.

---

> ### Author Response · Authors · 2024-11-20
> **Continued Response to Reviewer eFxS's Review**
>
> **Regarding the presentation of the manuscript:**
>
> - **Reviewer comment:** "Key concepts such as "deconvolution" or (presumably conjunction?) were not defined."
>
> 	**Response:** We have now formally defined "deconvolution" twice within the introduction:
>
> 		"Revealing MoAs alongside computational DD, a task we call MoA deconvolution"
>
> 		"We propose MoA deconvolution, the prediction of mechanistic pathways between drugs and their biological effects"
>
> 	We have also defined conjunction within Section 3.1:
>
> 		"...in which triples are connected by logical conjunctions (∧). Conjunctions indicate that, if all triples in the rule body are true, then the rule head is evaluated as true. "
>
> 	 Additionally, we note, in Section 3.3, when the definition of conjunction is relaxed:
>
> 		"We note two caveats. Firstly, to account for partial metapath matches, we relax our definition of conjunction here, allowing truth to be evaluated on the fragment level."
>
> 	 Finally, we defined other key concepts, including the following:
>
> 	 - "satisfied" is defined in Section 3.1: " If a rule head is evaluated as true, then the rule is satisfied"
>
> 	 - "rule body" is also explained in Section 3.1: " In MARS, metapaths are used as the bodies of logical rules, in which triples are connected by logical conjunctions (∧).
>
> 	 - "binary predicate" is explained in Section 3: " Here, we represent triples as binary predicates: for example, the binary predicate 'interacts(Protein, Protein)' states that two Protein nodes are connected via the interacts relation."
>
> - **Reviewer comment:** "The main predictive task was not explicitly formulated: from Figure 1A it appears to be link prediction. "
>
> 	**Response:** We now clarify this in the second paragraph of Section 3:
>
> 		"Specifically, we accomplish this through a link prediction task, in which we predict whether edges of type ‘induces’ exist between Drug and BP nodes. Thereafter, new predictions regarding the ‘induces’ relation serve as potential therapeutic outcomes for the chemical compound represented by the Drug node. "
>
> - **Reviewer comment(s):** "The background colour of Figure 1B makes it harder to read, plus the font sizes for the labels are a bit too small. " **+** "Moreover, it would be useful to have the catalogue of different entities--entities interaction types (Figure 1B) in a table."
>
> 	**Response:** We have decided to remove the old Figure 1 in favor of:
>
> 	 - Our new figure 2, which shows the MARS pipeline, and
> 	 -  Supplementary table A.5, as you suggested, denoting the number of entity and relation types.
>
> - **Reviewer comment:** " The citation for PoLo should use the published version (https://link.springer.com/chapter/10.1007/978-3-030-77385-4_22) "
>
> 	**Response:** Thanks for pointing this out. We have changed the citation accordingly.
>
> - **Reviewer comment:** "Could you include more training details e.g. hardware, training time, validation loss plots etc.?"
>
> 	**Response:** We have now included more details about training, including the time and memory requirements within Supplementary Sections A.2, "Implementation" and A.3, "Hardware and Resources".
>
> - **Reviewer comment:** " On what basis were the "feasible" MoA's chosen in Table A1? "
>
> 	**Response:** We have now clarified this within Section 3.4:
>
> 		"Using the hetnetpy package (Himmelstein et al., 2021), we extract all metapaths (see Section 3.1) from MoA-net which we considered to be valid MoAs: those comprising directed, mechanistic paths between drug and BP nodes (see Appendix A.1). We exclude metapaths depicting associative patterns, such as those leveraging information about shared BP targets, from our set of metapath-based rules. "
>
> 	Furthermore, we make the distinction between associative and mechanistic patterns clearer within our new Fig. 1 as well as the third paragraph of introduction:
>
> 		"Unfortunately, these associative patterns, which are discovered during or after clinical trials, are rare or absent for novel compounds. Furthermore, such patterns can not represent the MoA of a drug; instead, an MoA involves mechanistic patterns, such as the physical, molecular interactions shown in Fig. 1."

---

> ### Comment · Reviewer_eFxS · 2024-11-25
>
> Thanks to the authors for replying to the comments! The revised version has a much clearer presentation.
>
> There are still a few more outstanding issues:
> - For the "pruned" evaluations (Table A1), how was the maximum length of $ 4 $ chosen (`Based on MoAs found in DrugMechDB, we limit metapaths to a maximum length of four relations (or hops)`)? This is partially a follow-up to a question of Reviewer FcaQ. Perhaps it may be helpful to plot the distribution of chain length in DrugMechDB?
> - Similarly, just for clarification, is it the case that you discarded edge type (`Predicate`) and certain types of vertices (e.g. `ChemicalSubstance`, `Cell`) from DrugMechDB?
> - What was the motivation behind trimming the dataset? Was it because of memory overflow? Presumably the `90 Gigabytes of memory` ($ \S $A.3) usage was for the trimmed dataset given that training took only $ 1.5 $ hours?
> - Could you comment on why on the standard evaluation the proposed method was outperformed by MINERVA, which also outperformed PoLo?
> - In Figure 5, was early-stopping applied at $ \sim 90 $ epoch?
> - Hyperparameter optimization:
>     - What was the best combination?
>     - The number of values for some of the hyperparameters seems a bit low, especially for $ \alpha $ and $ \lambda $. Have more LSTM layers than $ 2 $ been tried? (If computational costs would have been a problem, why not try Bayesian optimization?)
>     - What was the meaning of $ \beta $, $ \gamma $ and $ \gamma_\mathrm{baseline} $ (presumably they are terms in the Bellman equation)?
>     - Which optimizer was used (presumably `Adam`)? Would it be a good idea to try a newer variant (e.g. `RAdam`)?
>     - The `search space` for `hidden size` is missing comas between the values.
> - The results for DWPC did not have any standard deviation - was it because it is a deterministic method?

---

> > ### Author Response · Authors · 2024-11-27
> > **Additional Response to Reviewer eFxS (Part 1)**
> >
> > Thanks to the reviewer for their thorough engagement in the review process! We are glad the paper is clearer, and we are happy to answer these additional questions below:
> >
> > - **Reviewer comment:** For the "pruned" evaluations (Table A1), how was the maximum length of 4 chosen (Based on MoAs found in DrugMechDB, we limit metapaths to a maximum length of four relations (or hops))? This is partially a follow-up to a question of Reviewer FcaQ. Perhaps it may be helpful to plot the distribution of chain length in DrugMechDB?
> >
> >     **Author response:** A maximum length of '4 hops' was chosen because it covers all of the MoAs occurring between drugs and BPs in DrugMechDB. We agree that including a plot of the chain length distribution would be helpful, so it is now included in Appendix A.2.
> >
> > - **Reviewer comment:** Similarly, just for clarification, is it the case that you discarded edge type (Predicate) and certain types of vertices (e.g. ChemicalSubstance, Cell) from DrugMechDB?
> >
> >     **Author response:** Regarding the ChemicalSubstance type, we used this type of entity if it corresponded to a drug in our KG. Often, the other types in DrugMechDB occur after the BP, such as drug -> protein -> BP -> phenotype -> disease, in which case we only took the part of the path up to the BP. In other cases, such types may also occur in between a drug and a BP. In that case, we did not consider these types when evaluating the correctness of a path. These processes are described in one of the code repositories included (MoA-net, exploration_drugmechdb.ipynb).
> >
> > - **Reviewer comment:** What was the motivation behind trimming the dataset? Was it because of memory overflow? Presumably the 90 Gigabytes of memory (§A.3) usage was for the trimmed dataset given that training took only hours?
> >
> >     **Author response:** We trimmed the dataset to test whether the approaches, (MARS_naive, MARS_P2H, and PoLo) could perform better when protein-protein connections were sparser. This is stated within Section 4.3:
> >
> >         "Next, we wanted to confirm that the agent was getting lost within the PPIs. As explained in Section 3.4, MoA-net-10k is a variant of MoA-net with fewer PPIs. We tested MARS_P2H , MARS_naive, and PoLo with the same parameters upon on MoA-net-10k"
> >
> >     Yes, the details discussed in (§A.3) are for MoA-net-10k.
> >
> > - **Reviewer comment:** In Figure 5, was early-stopping applied at epoch 90?
> >
> >     **Author response:** Early stopping was applied with a patience of 2 and evaluation every 10 epochs. Therefore, while MRR starts decreasing at epoch 90, training is not stopped until epoch 110.
> >
> > - **Reviewer comment:** (hyperparameter optimization) What was the best combination?
> >
> >     **Author response:** We now include the best hyperparameters in Table A4 of Section A.6.
> >
> > - **Reviewer comment:** The number of values for some of the hyperparameters seems a bit low, especially for alpha and and Lambda.
> >
> >     **Author response:** We are not convinced that alpha and Lambda are too low. In particular, Lambda balances the proportion of the supplementary reward to the base reward. Therefore, 5-10x, to us, seems quite high. As for some of the other hyperparameters, we chose their ranges based on what worked, previously, for PoLo: https://github.com/liu-yushan/PoLo/tree/main/configs .
> >
> > - **Reviewer comment:** Have more LSTM layers than 2 been tried?
> >
> >     **Author response:**  We have not tried more than 2 LSTM layers, but it could, in theory, be tuned.
> >
> > - **Reviewer comment:** (If computational costs would have been a problem, why not try Bayesian optimization?)
> >
> >     **Author response:**  As demonstrated by the reviewer's questions about the scope of each hyperparameter, the hyperparameter search space is high-dimensional - we have 15 hyperparameters and a continuous range of options for several of them. Since a known limitation of bayesian HP optimization is scalability to many parameters, we felt it simpler to choose hyperparameter options based upon the need of our study and previous work (https://github.com/liu-yushan/PoLo/tree/main/configs).

---

> > > ### Author Response · Authors · 2024-11-27
> > > **Additional Response to Reviewer eFxS (Part 2)**
> > >
> > > - **Reviewer comment:** What was the meaning of beta, gamma, and gamma baseline (presumably they are terms in the Bellman equation)?
> > >
> > >     **Author response:**  Thanks for pointing these out:
> > >
> > >     - Beta is the entropy regularization factor, as implemented by MINERVA (https://arxiv.org/pdf/1711.05851).
> > >
> > >     - Gamma is is the discount factor, as implemented in REINFORCE (so this one is as in the Bellman equation).
> > >
> > >     - Gamma baseline is the baseline discount factor, as implemented in MINERVA (https://arxiv.org/pdf/1711.05851). They implement a moving average of the cumulative discounted reward as a baseline for REINFORCE. This discount factor controls the weight of that baseline. Note that in MINERVA and PoLo, this parameter is called 'Lambda.' However, the authors of PoLo accidentally introduced another 'Lambda' parameter, which controls the balance between the supplementary and base rewards. We kept the latter definition for Lambda and renamed the former, 'gamma_baseline'
> > >
> > >     We now clarify the above in Tables A2 and A3.
> > >
> > > - **Reviewer comment:** Which optimizer was used (presumably Adam)? Would it be a good idea to try a newer variant (e.g. RAdam)?
> > >
> > >     **Author response:**  Yes, we used Adam, which we've now clarified and cited within Appendex A.2 Implementation. Using RAdam is also a good idea, but we used Adam to keep our approach more comparable to its predecessors, PoLo and MINERVA.
> > >
> > > - **Reviewer comment:** The search space for hidden size is missing comas between the values.
> > >
> > >     **Author response:**  Thanks for pointing this out- we've fixed it now.
> > >
> > > - **Reviewer comment:** The results for DWPC did not have any standard deviation - was it because it is a deterministic method?
> > >
> > >     **Author response:**  Yes, DWPC is a deterministic method. We've now clarified this within the caption of Table 1.
> > >
> > > Thanks again for all of the constructive feedback.

---

### Author Response · Authors · 2024-11-27
**Thanks to the reviewers**

Dear reviewers,

Thank you for your constructive feedback. We greatly appreciate all of the points and suggestions made, and we believe your reviews have helped us to improve our manuscript.

Please let us know if you have any further questions during this discussion period.

Thank you!

Sincerely,
the authors.

---

### Meta-Review · Area_Chair_sktD · 2024-12-19

**Metareview:**

This paper introduces a neurosymbolic system for drug discovery, which consists of drug mechanism-of-action (MoA) deconvolution; MoA-net and its variants, a specifically designed knowledge graph based on public biomedical data; MARS, a Neural Symbolic approach through dynamic rule-weight updates. The paper is about AI for science (biology and drug discovery), neurosymbolic approaches have the potential for interpretable drug discovery, and the paper is clearly written. However, reviewers raised concerns that the experiments conducted on a dataset don't sufficiently capture the biochemical complexity. For example, there is no differentiation between kinds of proteins (e.g. receptors, enzymes, transporters), drug-protein interactions should be more than just upregulates and downregulates, and drug acts by binding to multiple target proteins that are jointly responsible for a biological process. Thus, there lack of concrete case studies to demonstrate its real-world value in drug discovery.

**Additional Comments On Reviewer Discussion:**

There are detailed discussions between the reviewers and authors. Since this paper is really a borderline one, the AC calls the discussions among reviewers and AC. There is no reviewer championing the paper. Some reviewers questioned innovation from an ML perspective and some questioned the biochemical complexity from a biology perspective.

---

### Decision · Program_Chairs · 2025-01-22

Reject